# Entangled polymer dynamics beyond reptation

Maram Abadi [1], Maged F. Serag [1] & Satoshi Habuchi [1]

Macroscopic properties of polymers arise from microscopic entanglement of polymer chains. Entangled polymer dynamics have been described theoretically by time- and space-averaged relaxation modes of single chains occurring at different time and length scales. However, theoretical and experimental studies along this framework provide oversimplified picture of spatiotemporally heterogeneous polymer dynamics. Characterization of entangled polymer dynamics beyond this paradigm requires a method that enables to capture motion and relaxation occurring in real space at different length and time scales. Here we develop new single-molecule characterization platform by combining super-resolution fluorescence imaging and recently developed single-molecule tracking method, cumulative-area tracking, which enables to quantify the chain motion in the length and time scale of nanometres to micrometres and milliseconds to minutes. Using linear and cyclic dsDNA molecules as model systems, our new method reveals chain-position-dependent motion of the entangled linear chains, which is beyond the scope of current theoretical framework.

[1] Biological and Environmental Sciences and Engineering Division, King Abdullah University of Science and Technology (KAUST), Thuwal 23955-6900, Saudi Arabia. Correspondence and requests for materials should be addressed to S.H. (email: Satoshi.Habuchi@kaust.edu.sa)

Rheological properties of polymer materials microscopically arise from entanglement of polymer chains[1–3]. Decades of theoretical, experimental[4–8], and simulation studies[9–11] suggested that topological states of polymer (e.g. linear, cyclic, etc.) have significant effect on entanglement mode between the chains[12]. Recent studies demonstrated that topological states of polymer chain is one of the key factors that regulate macroscopic physical properties of polymer materials, including thermal stability of polymer micelles[13,14] and elastic properties of polymer gels[15]. Nanoscopic characterisation of the entanglement between topological polymers at the single-chain level would thus provide a foundation for the development of new polymer materials.

Motion of a linear polymer chain under entangled conditions has been described by reptation theory[2]. In this theory, a polymer chain is confined in a transiently existing virtual tube created by entangled surrounding chains. Due to this spatial confinement, the chain cannot move transversely across the tube and displays motion only along the tube. According to this model, motion of the entire chain is determined by the motion of two ends of the chain. It is obvious that the reptation theory cannot fully describe the motion of topologically unique cyclic polymer chains under entangled conditions as cyclic chains do not have chain end. Several theoretical frameworks describing motion and relaxation of entangled cyclic chains have been proposed, including double-folded-reptation, lattice-animal, constraint release, and once-threaded model[16–20]. Contribution of mutual relaxation between entangled chains and decoupling of diffusive motion and chain relaxation on entangled cyclic polymer dynamics have also been suggested[21,22]. Compared with well-accepted reptation model for linear polymers, cyclic polymer dynamics remains elusive.

Nuclear magnetic resonance spectroscopy, light and neutron scattering, and viscosity and stress-relaxation measurement have been main tools for characterising polymer dynamics[23–25]. Although most of the current polymer physics theories have been established based on the findings obtained using these methods, characterisation of entangled polymer dynamics at the single-chain level is impossible with these ensemble-averaged experimental methods. Single-molecule techniques provide a tool for direct visualisation and characterisation of motion and relaxation of single chains under entangled conditions. Reptation model has been confirmed for linear chains using natural polymers such as DNA[26] and actin filament[27] as well as synthetic polymers[28]. In these studies, although motion and conformational state of single chain were captured directly, quantitative analyses were often conducted based on overall motion and relaxation of the chains such as chain-length-dependent motion of centre of mass (CM). Since reptation theory, in principle, describes quantitatively time- and space-averaged relaxation modes of single chains occurring at different time- and length scales, such scaling laws analyses are necessary to describe the motion and relaxation at the single-chain level based on this theory[28–32]. Although the scaling law analyses provide useful tool to confirm reptation theory, complete characterisation of local and whole chain motion of single molecules in real space beyond the theory is impossible with these frequently used approaches. The real-space subchain-level analysis would be particularly important to characterise motion of topological polymers, including cyclic polymers, as theoretical framework to describe their entangled dynamics has not been established. While subchain-level analyses of the conformational dynamics of spatially isolated rigid biopolymers have been reported[33–35], nanoscopic characterisation of entanglement between flexible polymer chains remain challenging.

Here we report visualisation and real-space characterisation of both local and global motion of single polymer chains under entangled conditions using super-resolution (SR) fluorescence localisation microscopy[36] and cumulative-area (CA) tracking[37], a single-molecule tracking algorithm that we recently developed. Using fluorescently labelled linear and cyclic double-stranded DNA (dsDNA; contour lengths of 16.5 and 14 μm, respectively) in semi-dilute solution of linear dsDNA as model systems, we visualise temporal dynamics of the single chains with 33 nm spatial resolution. We characterise the motion at whole chain level by an accurate determination of contours of the molecule with 6–33 nm precision. Timescale and amplitude of local motion of entangled segments along the contours is characterised quantitatively by estimating the amplitude of local motion perpendicular to the contours by combining SR localisation microscopy and CA tracking, and by the widths of the localisations at each position along the contours. While the motion of the linear chain is qualitatively consistent with the reptation theory, our analysis reveals chain position-dependent motion of the linear molecule, which is beyond the scope of the reptation theory. We also capture theoretically predicted but unproven conformational states of the cyclic chains with modes of the motion distinct from those of the linear chains.

## Results

**Evaluation of the experimental system.** We used lambda DNA (48.5 kbp, contour length of 16.5 μm) and Charomid 9-42 DNA (42 kbp, contour length of 14 μm) as model polymers for linear and cyclic chains. Given its well-characterised elastic properties in entangled conditions (i.e. reptation motion is suggested by scaling law behaviours[32]), linear DNA molecules serve as an excellent reference system for entangled polymer dynamics. These DNA molecules were fluorescently labelled by Cy5 dyes. Unlike conventional labelling with intercalator dyes, Cy5 dyes were covalently attached to heteroatoms on DNA molecule through a flexible linker (Fig. 1a)[38]. This allowed fluorescence labelling of DNA with minimum effect on its structure and motion under entangled conditions. The DNA samples were labelled at relatively high density (5–15 bp per dye). Since the Kuhn length of dsDNA is approximately 100 nm[39], the DNA molecules used in this study consist of 140–165 Kuhn monomer, which can be treated as semi-flexible polymers. The fluorescently labelled tracer DNA molecules were mixed with a semi-dilute solution of matrix DNA (non-labelled lambda DNA at 5–10 mg ml$^{-1}$ concentration)[21]. Details of the sample preparation are described in the Methods.

We visualised nanoscopic conformations of individual tracer chains using SR fluorescence localisation microscopy. Prior to capturing images in the entangled solution, we recorded SR fluorescence images of Cy5-labelled DNA deposited on a glass surface to evaluate the image resolution obtained for our samples (see Methods for details). Conventional fluorescence image obtained using epi-fluorescence microscopy displayed diffraction-limited width (approximately 300 nm) of the deposited DNA molecule (Fig. 1b). On the other hand, reconstructed SR image of the same molecule showed the width of approximately 30 nm (Fig. 1c, d). This result demonstrates that our labelling and image acquisition scheme allows us to capture images with an order of magnitude higher resolution compared with that obtained by conventional fluorescence microscopy, which enables us to capture much more detailed conformational state of the molecule.

To capture real-time motion of the tracer chains in three-dimensional (3D) space in the matrix solution using SR localisation microscopy, fast image acquisition speed, 3D resolution, and long-term mechanical stability of the microscope were necessary (Fig. 2a, see Methods). Image acquisition speed was increased to 1–1.4 kHz by proper masking and cropping of the images captured by electron multiplying charge-coupled

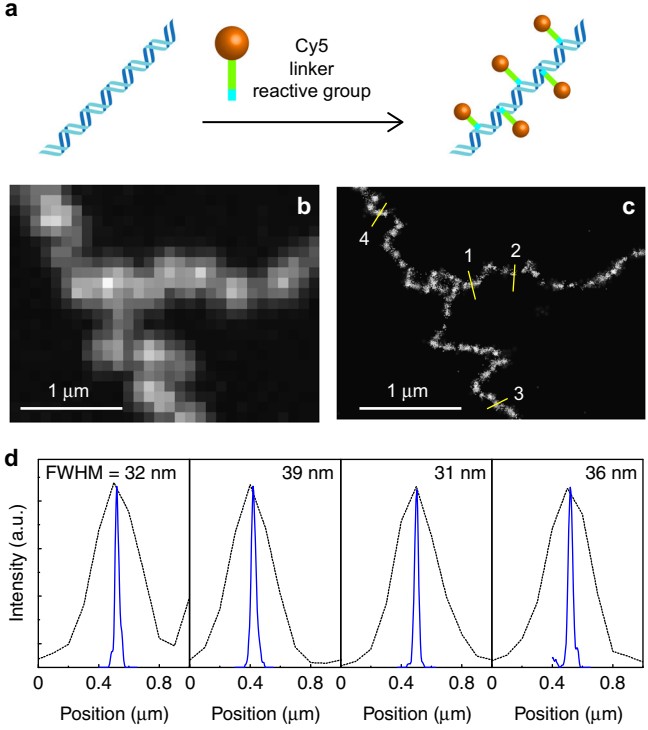

**Fig. 1** Super-resolution fluorescence imaging of DNA using covalently conjugated fluorophores. **a** Schematic illustration describing covalent conjugation of dsDNA to Cy5 dyes. **b** Conventional fluorescence image of Cy5-labelled lambda DNA deposited on a glass surface. **c** Super-resolution fluorescence localisation microscopy image of the Cy5-conjugated DNA displayed in **b**. **d** Fluorescence intensity profiles obtained at four different positions of the molecule displayed in **c**. The dotted lines show fluorescence intensity profiles obtained from the conventional fluorescence image displayed in **b**

device (EM-CCD) camera. With this data acquisition speed, we were able to reconstruct SR images with 7–10 s image acquisition time (we reconstructed each SR image using 10,000 frames). Long-term motion of the molecules was capture by recording 10,000 frames of fluorescence images every half to two minutes. A stage drift along the axial direction was controlled by a focus correction system. We used astigmatism-based 3D SR localisation microscopy[40] to reconstruct 3D images of the tracer molecules by inserting a cylindrical lens in front of the EM-CCD camera.

Conventional fluorescence image of a linear tracer chain in the entangled semi-dilute solution (Fig. 2b) shows elongated shape of the molecule that corresponds to the snapshot of the conformational state of the molecule. We then switched to the SR localisation microscopy mode, captured 10,000 frames of raw images in 7 s, and reconstructed SR image (Fig. 2c). While the conformation of the molecule is captured with much higher image resolution in the SR mode, two images show perfect overlap. This result confirms that the molecule does not show motion at the entire chain level such as reptation motion during the 7 s image acquisition time (see below for the local motion occurring during the image acquisition) and suggest that we can capture a snapshot of the conformational state of the molecule using our SR fluorescence microscopy technique. Time-lapse SR images of the linear tracer chain in the semi-dilute solution (Fig. 2d) captured every 1 min clearly show the motion at the whole chain level. Total data acquisition time is limited typically to 7–12 min by several factors, including photobleaching of the labelled Cy5 dyes, degradation of switching buffer, and long-term

mechanical stability of the experimental set-up. Nevertheless, the data displayed in Fig. 2d demonstrate that we are able to visualise nanoscopic motion of the entangled polymers by the time-lapse SR imaging.

**Analytical tools of the chain motion under entangled conditions**. We characterised the motion of the molecule at the entire chain level by an accurate determination of 3D contours of the chain. More quantitative analyses were conducted using two-dimensional (2D) contours of the chain projected on $XY$ plane (see Methods for details). We first determined local directions of the contour by fitting the localisations in small segments (the size of each segment was 100 nm) by piecewise linear function. Then, local centroids were determined by fitting the localisations in small segments by one-dimensional (1D) Gaussian function along the direction perpendicular to the local directions of the contour. The 3D contour of the molecule is determined by connecting nearest neighbouring centroids (Fig. 3a). The 2D contour of the molecule is determined in a manner similar to the 3D contour, but using 2D coordinate of the localisations in the reconstructed image (Fig. 3b). Mean standard deviations of the Gaussian ($\sigma_{av}$) obtained from the fitting was approximately 14 nm (Fig. 3c). Thus, the image resolution in this experiment defined by the full width at half maximum of the Gaussian is estimated to be 33 nm. The peaks of the Gaussian are determined with 6 nm precision (Fig. 3d). Although this result does not mean that we determined the contour with 6 nm precision because the segment size for the Gaussian fitting is much larger than this value and because of the local motion of the chain during the 7 s image acquisition time (see below for details), the data shown in Fig. 3c, d suggest that we determine the contours of the chain with 6–33 nm precision. By analysing displacement between contours determined for consecutive reconstructed SR images, we characterised mode of the motion of entangled chains occurring at the timescale of minutes.

Local motion of the tracer chain was quantified by two different methods (see Methods for details). First, we drew the contour of the chain with the widths (i.e. standard deviations) of the Gaussian obtained for each segment (Fig. 3e). The width of the Gaussian is determined by localisation precision of each fluorescent spot and local motion of the chain along the direction perpendicular to the contour. Since the former contribution is determined by the number of photon detected in each localised spot that is independent on the positions along the contour of the chain[41], the latter determines the position-dependent width of the Gaussian when mean widths of the Gaussian at each position are calculated from a large number of molecules. By analysing the local widths along the chain, we characterised the amplitude of the local motion of the chain that occurs in the time scale of capturing one SR image (7–10 s) in real space.

Second, we applied CA tracking to each segment. CA tracking is a single-molecule tracking algorithm developed recently in our group (see Methods for details)[37]. In this method, diffusion speed of a single molecule is determined by time-dependent increase of CA occupied by the moving molecule. In this study, we modified the original CA tracking and calculated the time-dependent increase of the CA based on the time-dependent spreading of the localised spots in each segment (Fig. 3b, f, g). Stochastic nature of the spatiotemporal distribution of the localisations and the actual motion of each segment contribute to the time-dependent spreading of the localised spots. The former contribution was evaluated by analysing the surface-deposited molecule (Fig. 3h), from which mean characteristic time of a few seconds was obtained. The characteristic times of 0.31 and 3.7 s can, therefore, be attributed to the local motion and spatiotemporal distribution of the localisations, respectively (Fig. 3g). By analysing the

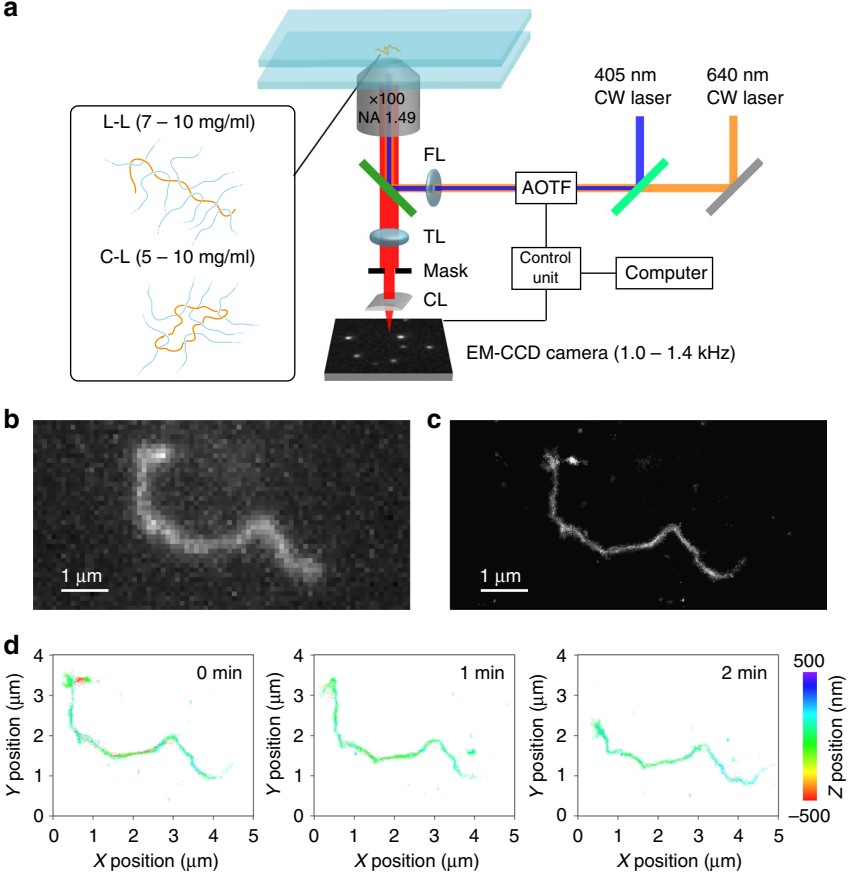

**Fig. 2** Time-lapse 3D super-resolution fluorescence imaging of entangled lambda DNA. **a** Schematic illustration describing the experimental configuration. AOTF acousto-optic tunable filter, FL focusing lens, FC focus correction system, TL tube lens, CL cylindrical lens. **b** Conventional fluorescence image of Cy5-labelled lambda DNA in an aqueous solution containing 10 mg ml$^{-1}$ concentration of non-labelled lambda DNA. **c** 2D projected super-resolution fluorescence image of the DNA molecule displayed in **b**. The super-resolution image was reconstructed from 10,000 frames of images captured at 1 kHz frame rate. **d** Time-lapse 3D super-resolution fluorescence images. Each super-resolution image was reconstructed from 10,000 frames of images captured at 1 kHz frame rate. The time-lapse super-resolution fluorescence images were obtained by recording the 10,000 frames of images every 1 min. The colour bar shows the axial position of the localised spots

characteristic times at each segment along the chain, we quantified the timescale of the local motion occurring in the timescale of capturing one SR image (7–10 s) in real space (Fig. 3g).

**Reptation and constraint release motion of linear chains**. The 3D contours of the tracer linear chain in entangled semi-dilute solution (10 mg ml$^{-1}$) show the motion of the molecule occurring in 10 min timescale (Fig. 4a). A frequency histogram of the contour lengths shows a broad distribution with maximum length close to the contour length of the molecule (Fig. 4b). This result suggests that while we captured primitive paths of the chains rather than actual contours of the chains, the primitive path length of the chains approaches to their contour length due to the high level of spatial confinement of the chains at the concentrations above the entanglement concentration ($C_e$) of lambda DNA ($C_e \approx 0.6$ mg ml$^{-1}$)[32] (see below). The contours of the chain obtained in two consecutive time frames (i.e. motion occurring in 1 or 2 min, Fig. 4c, d) clearly demonstrate that the chain shows motion along the contour (tube-like motion, Fig. 4c, d, white squares). Reptation motion is characterised by a confined motion of polymer chains along virtual tube, in which the tube diameter is one of the key parameters that characterises reptation motion[42,43]. Given the concentration of the semi-dilute solution

(10 mg ml$^{-1}$) and the scaling law of reptation model[44,45], the tracer linear chain has 158 entanglements with the matrix chains, which correspond to the entanglement length and the tube diameter of approximately 51–95 nm (see Methods for detailed calculation). The displacement of the chain between the consecutive time frames along the direction perpendicular to the contour at the centre region of the chain was less than the diameter of the tube (Fig. 4c, d, red squares), consistent with the prediction by the reptation theory. A similar behaviour was observed for the linear chain in slightly lower 7 mg ml$^{-1}$ concentration of the semi-dilute solution. Under this condition, the tracer chain has 100 entanglements with the matrix chains, which correspond to the entanglement length and the tube diameter of approximately 80–150 nm (Fig. 4e, see Methods for detailed calculation).

The chain ends showed large motional freedom compared with the rest of the chain (Fig. 4c, d, green squares), which corresponds to the creation of new tube at the chain ends, consistent with the reptation model. Occasionally, we observed small chain regions that exhibited the displacement larger than the tube diameter (Fig. 4c, d, cyan squares). This observation indicates that constraint release motion, in which the constraints of the chain due to the entanglement is released through motions of the surrounding chains[46], is operative. According to the previous study on entangled DNA[32], the entanglement molecular weight

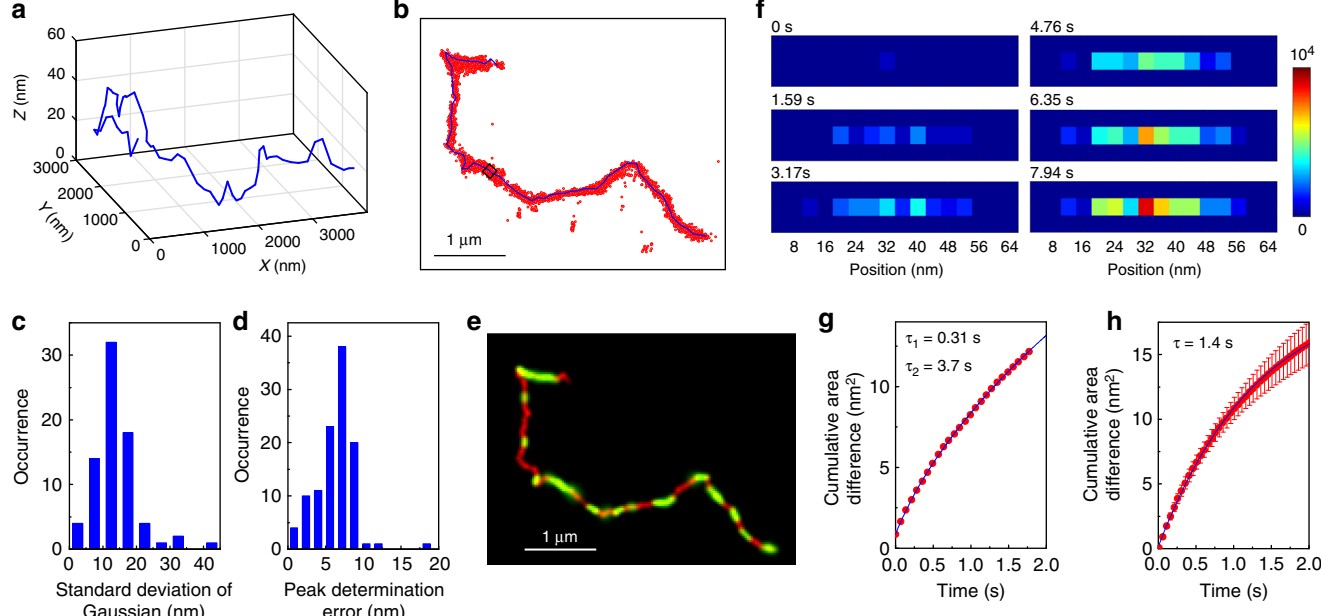

**Fig. 3** Characterisation of local and global motion of single entangled DNA molecules. **a** 3D contour of the DNA molecule displayed in Fig. 2d. **b** 2D projected contour of the DNA molecule shown in **a**. The red dots show the localised spots determined in the super-resolution fluorescence imaging experiment. The black square shows the region from which we obtained cumulative-area images displayed in **f**. **c** Frequency histogram of the standard deviation of Gaussian obtained by the fitting of the local intensity profiles of the reconstructed super-resolution images at each segment. **d** Frequency histogram of peak determination error in the 1D Gaussian fitting of the local intensity profiles of the reconstructed super-resolution images at each segment. **e** 2D projected contour of the DNA molecule displayed in Fig. 2c. The local widths of the contour correspond to the standard deviations of the Gaussian at each local segment. The regions that show standard deviation of the Gaussian smaller than that obtained from the super-resolution imaging of the surface-deposited molecule are shown in red, in which the width of the Gaussian is determined by localisation error rather than local motion of the molecule. **f** Cumulative-area images obtained from the local region shown in **b** at time 0, 1.59, 3.17, 4.76, 6.35, and 7.94 s. **g** Cumulative-area difference plots obtained from the local region shown in **b** (top). Characteristic times were calculated by fitting the data to double-exponential function (solid line). **h** Mean cumulative-area difference plot obtained from 10 local segments of surface-deposited DNA molecules. Error bars correspond to the standard deviations. Characteristic time was calculated by fitting the data to single-exponential function (solid line)

($M_e$) of DNA is estimated to be between 9.4 and 23.1 kbp. The molecular weight of lambda DNA (48.5 kbp) is therefore 2.1–5.1 times larger than $M_e$. Previous studies on synthetic polymers in melts suggested that the constraint release motion influences self-diffusion of entangled polymer when the molecular weight of the polymer ($M_w$) is in the range between $M_e < M_w < 10M_e$[47–49]. Therefore, the local chain displacements larger than the tube diameter observed in this study can be interpreted reasonably by the constraint release motion of the chain. To the best of our knowledge, we captured for the first time unambiguous sign of the constraint release motion of polymer chains. At the moment, we cannot distinguish distinct constraint release models proposed previously[16,46,50] based on our observation. Nevertheless, these results strongly suggest that our imaging platform enables unprecedented subchain-level characterisation of entangled polymer dynamics in real space.

**Linear-chain motion beyond reptation theory**. Reptation motion is defined by several characteristic timescales, including entanglement time (i.e. Rouse time of the entangled strand, $\tau_e$), Rouse rotational relaxation time (i.e. characteristic time during which a polymer molecule diffuses over a distance of its size, $\tau_r$), and disentanglement time (i.e. characteristic time during which a polymer chain moves out of its original tube, $\tau_d$)[51]. Thus, we calculated time- and length scales of the chain motion characterised by analysing displacement between contours determined for consecutive reconstructed SR images and by analysing the local widths of the Gaussian and the characteristic times along the chain and compared them with values predicted by the reptation

model for quantitative analysis of the entangled polymer dynamics.

The analysis of the displacement between the contours obtained for adjacent time points revealed that the local chain displacement along the direction perpendicular to the contour increases towards the chain ends and displays displacement larger than the tube diameter at the chain ends (Fig. 5a, see Methods for details of the analysis). Similar position-dependent displacement was observed when the analysis was conducted using the 3D contours of the molecules (Fig. 5b, see Methods for details of the analysis), further supporting the validity of the observation. A large motional freedom at the chain ends has been captured directly in previous studies using synthetic polymers[28,42]. In these studies, the large motional freedom at the chain ends was observed in the time scale of reptation motion (i.e. $\tau_r < t < \tau_d$), which can be interpreted within the framework of the reptation theory. In this study, we observed the chain position-dependent motion at the timescale of minute, which is much shorter than the estimated Rouse rotational relaxation time ($\tau_r$) of lambda DNA at 7–10 mg ml$^{-1}$ concentration (1.2–3.3 h, see Methods for detailed calculation), suggesting that the chain position-dependent motion observed in this study is not directly related to the reptation motion of the chain. Instead, given 158 entanglements per chain, the gradual decrease of the local chain displacement towards the centre of the molecule (Fig. 5a, b) indicates that the confinement level in the tube changes gradually over tens of entanglements from the chain ends. Although a radial position-dependent relaxation has been reported for linear DNA molecules in a dilute solution[52], the observed position-

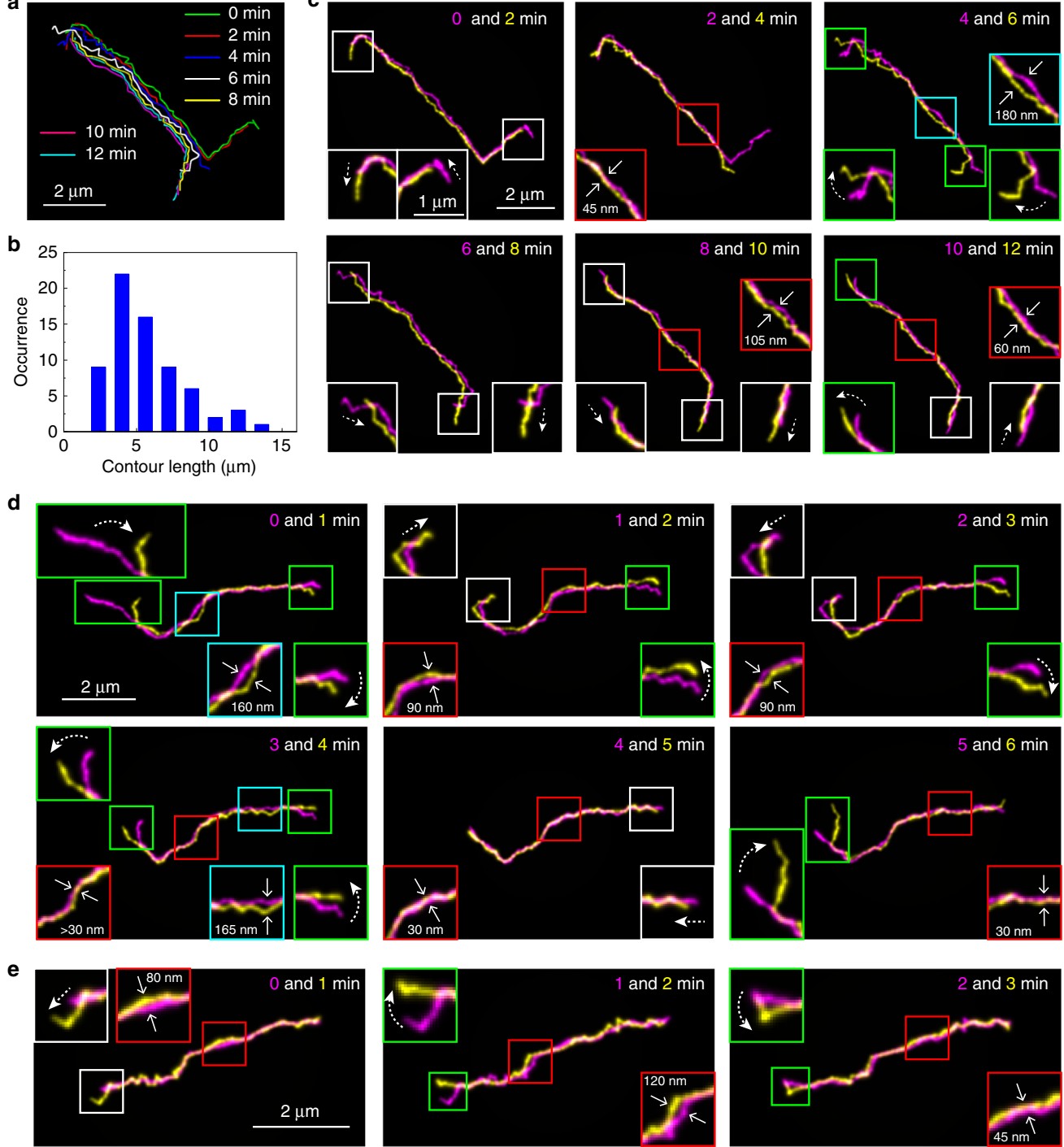

**Fig. 4** Motion of the linear DNA molecules under entangled conditions. **a** Superimposed 3D contours of the Cy5-labelled lambda DNA in an aqueous solution containing 10 mg ml$^{-1}$ concentration of non-labelled lambda DNA. **b** Frequency histogram of the contour lengths of Cy5-labelled lambda DNA determined by the 3D contours of the molecules. The contours of the molecules were obtained in an aqueous solution containing 7–10 mg ml$^{-1}$ concentration of non-labelled lambda DNA. **c** 2D projected contours of the DNA molecule displayed in **a**. The local widths of the contours correspond to the standard deviations of the Gaussian at each local segment. Two contours obtained from adjacent time frames are shown in each panel in cyan (earlier time points) and yellow (later time points). Insets show enlarged views of the areas highlighted by the squares. **d** Another example of 2D projected contours of a Cy5-labelled lambda DNA molecule in an aqueous solution containing 10 mg ml$^{-1}$ concentration of non-labelled lambda DNA captured at different time points. The local widths of the contours correspond to the standard deviations of the Gaussian at each local segment. Two contours obtained from adjacent time frames are shown in each panel in cyan (earlier time points) and yellow (later time points). Insets show enlarged views of the areas highlighted by the squares. **e** 2D projected contours of a Cy5-labelled lambda DNA in an aqueous solution containing 7 mg ml$^{-1}$ concentration of non-labelled lambda DNA. The local widths of the contours correspond to the standard deviations of the Gaussian at each local segment. Two contours obtained from adjacent time frames are shown in each panel in cyan (earlier time points) and yellow (later time points). Insets show enlarged views of the areas highlighted by the squares

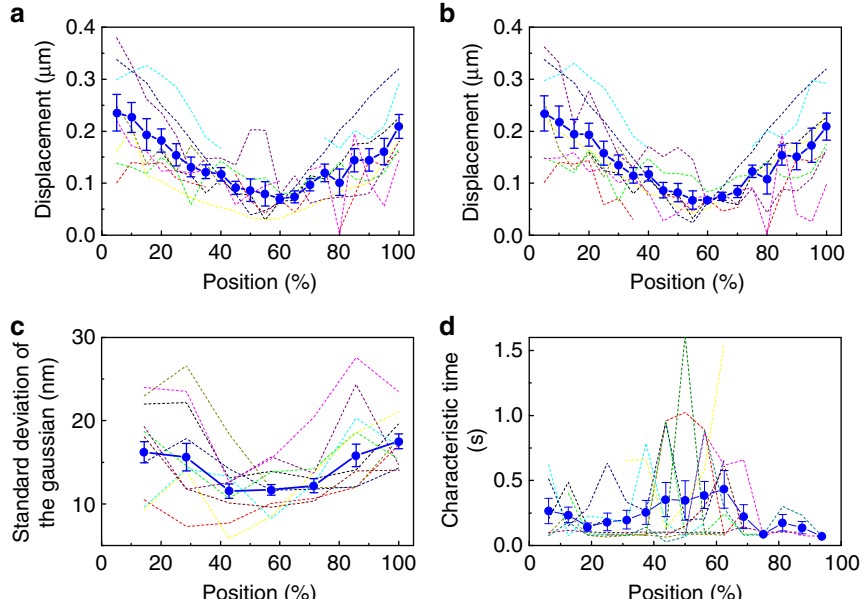

**Fig. 5** Quantitative analysis of the motion of the entangled linear DNA molecules. **a**, **b** Mean position-dependent displacement of the Cy5-labelled lambda DNA occurring along the direction perpendicular to the contours during the time gap between the adjacent time frames in an aqueous solution containing 7–10 mg ml$^{-1}$ concentration of non-labelled lambda DNA. The displacements were calculated using **a** 3D contours and **b** 2D projected contours of the DNA molecules. The error bars show standard error of the means determined using 9 independent images of the DNA molecules. The dotted lines show the position-dependent displacement obtained from the 9 independent images of the DNA molecules. **c** Mean position-dependent standard deviation of the Gaussian obtained from the Cy5-labelled lambda DNA in an aqueous solution containing 7–10 mg ml$^{-1}$ concentration of non-labelled lambda DNA by the fitting of the local intensity profiles of the reconstructed super-resolution images at each segment. The error bars show standard error of the means determined using 15 independent images of the DNA molecules. The dotted lines show the position-dependent standard deviation of the Gaussian obtained from the 15 independent images of the DNA molecules. **d** Mean position-dependent characteristic time obtained from the Cy5-labelled lambda DNA in an aqueous solution containing 7–10 mg ml$^{-1}$ concentration of non-labelled lambda DNA by the fitting of the cumulative-area difference plots at each segment. The error bars show standard error of the means determined using 12 independent images of the DNA molecules. The dotted lines show the position-dependent characteristic time obtained from the 12 independent images of the DNA molecules

dependent chain confinement under entangled conditions cannot be interpreted by current polymer physics theories.

The position-dependent chain motion was further characterised by analysing the amplitude and timescale of the local motion occurring in the timescale of capturing one SR image (7–10 s). The mean width of the Gaussian (standard deviation of 14 nm, Fig. 3c, e) corresponds to the $1/e^2$ width of 60 nm, which is close to the tube diameter. The mean characteristic time obtained by the CA tracking analysis (0.31 s, Figs. 3g and 5d) agree well with the entanglement time ($\tau_e$) estimated by the reptation model[51] ($\tau_e = 0.47$ s, see Methods for detailed calculation). These results suggest that the amplitude and timescale of the local motion quantified by this analysis capture the local motion of the entangled strands confined in the tube predicted by the reptation theory in a direct way without using scaling law analyses. While the mean amplitude and timescale of the local motion can be interpreted by the reptation model, the position-dependent width of the Gaussian determined for multiple molecules clearly showed wider widths near the chain ends compared with the centre part of the chain (Figs. 4c, d and 5c, see Methods for details of the analysis). The position-dependent CA tracking analysis also shows gradual decrease of the characteristic time towards the chain ends. Our results suggest that the larger and faster motion of entangle segments towards the chain ends is responsible for the gradual change in the confinement level of the chain in the tube. Mutual relaxation of the entangled chains, which is not considered in the reptation theory, might be responsible for this behaviour.

Our analysis revealed the position-dependent motion of the chain, which is beyond the scope of the reptation theory that

describes time- and space-averaged relaxation modes of single chains occurring at different time- and length scales. These results also demonstrate that the combination of SR fluorescence localisation microscopy and CA tracking analysis enables one to access a wide range of space- and temporal resolution that cannot be achieved by conventional single-molecule techniques and thus offers unprecedented opportunities to characterise polymer dynamics occurring at the level of entangled strands in real space.

**Motion and conformation of cyclic chains governed by topological constraint**. Motion of cyclic chains under entangled conditions captured by the SR imaging experiment was distinct from that of the linear chains (Figs. 6 and 7). A frequency histogram of the contour lengths shows a broad distribution with maximum length close to the contour length of the molecule (Fig. 6d), suggesting that we captured primitive path length of the chains[11,20,53], similar to the linear chains. The time-lapse images captured under the experimental condition identical to that for the linear chains (i.e. concentration of the matrix chains of 10 mg ml$^{-1}$) are characterised by small and non-directional motion (Fig. 6a, b). This is in contrast to the linear molecule (Fig. 4c, d) and agrees well with previous observation that the circular DNA molecule under the entangled conditions shows slower and non-directional diffusional motion at the length scale of the molecule[21]. The contours of the chain obtained in two consecutive time frames displayed only non-directional position-independent displacement (Fig. 6b), indicating a deviation from the reptation motion. The quantitative analysis of the chain displacement shows that the cyclic chains do not show any position-dependent displacement of the chain along the

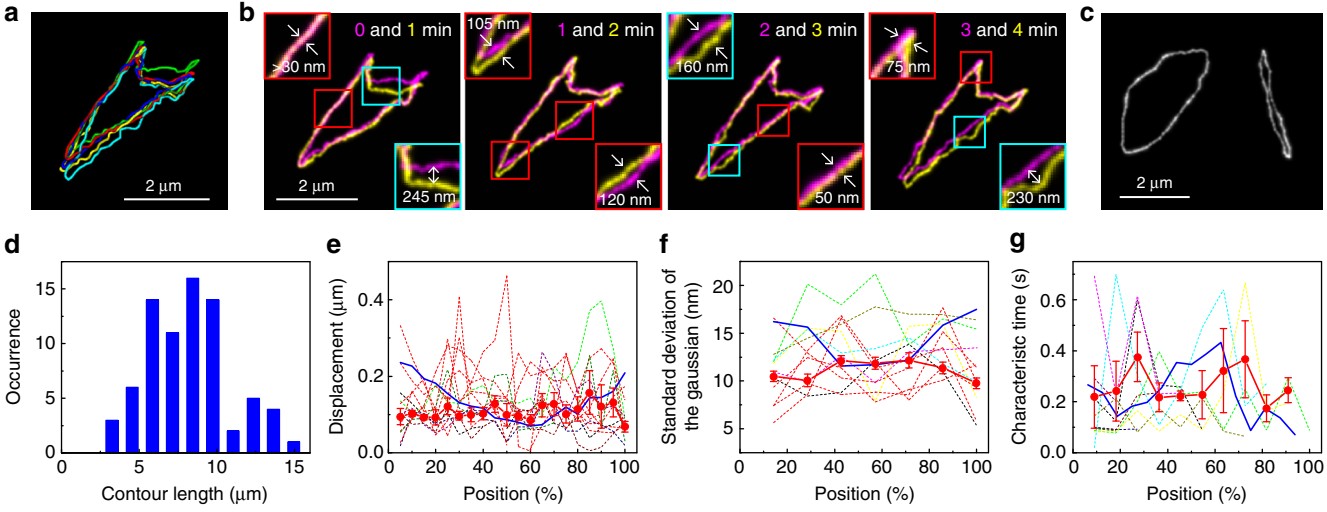

**Fig. 6** Analysis of the motion of the entangled cyclic DNA molecules. **a** Superimposed 3D contours of the Cy5-labelled Charomid 9-42 DNA in an aqueous solution containing 10 mg ml$^{-1}$ concentration of non-labelled lambda DNA. **b** 2D projected contours of the DNA molecule displayed in **a**. The local widths of the contours correspond to the standard deviations of the Gaussian at each local segment. Two contours obtained from adjacent time frames are shown in each panel in cyan and yellow. Insets show enlarged views of the areas highlighted by the squares. **c** Examples of the 2D projected contours of the Cy5-labelled Charomid 9-42 DNA in an aqueous solution containing 10 mg ml$^{-1}$ concentration of non-labelled lambda DNA. **d** Frequency histogram of the contour lengths of Cy5-labelled Charomid 9-42 DNA determined by the 3D contours of the molecules. The contours of the molecules were obtained in an aqueous solution containing 7–10 mg ml$^{-1}$ concentration of non-labelled lambda DNA. **e** Mean position-dependent displacement of the Cy5-labelled Charomid 9-42 DNA occurring along the direction perpendicular to the contours during the time gap between the adjacent time frames in an aqueous solution containing 7–10 mg ml$^{-1}$ concentration of non-labelled lambda DNA (red line). **f** Mean position-dependent standard deviation of the Gaussian obtained from the Cy5-labelled Charomid 9-42 DNA in an aqueous solution containing 7–10 mg ml$^{-1}$ concentration of non-labelled lambda DNA by the fitting of the local intensity profiles of the reconstructed super-resolution images at each segment (red line). **g** Mean position-dependent characteristic time obtained from the Cy5-labelled Charomid 9-42 DNA in an aqueous solution containing 7–10 mg ml$^{-1}$ concentration of non-labelled lambda DNA by the fitting of the cumulative-area difference plots at each segment (red line). The error bars in **e**–**g** show standard error of the means determined using 12, 11, and 10 independent images of the DNA molecules, respectively. The dotted lines in **e**–**g** show the data obtained from the 12, 11, and 10 independent images of the DNA molecules. The blue line shows the data obtained from lambda DNA

direction perpendicular to the contour (Fig. 6e, see Methods for details of the analysis).

While the overall motion of the cyclic chain is distinct from that of the linear chain, they showed similar local motion. The displacement of the cyclic chain between the consecutive time frames along the direction perpendicular to the contour is smaller than the tube diameter of the confined linear chain (Fig. 6b, red squares) with occasional deviations (Fig. 6b, cyan squares). These observations suggest that the chain interactions at the local level in both linear and cyclic chains are described by the same mechanism based on the spatial confinement of the molecule by the surrounding chains and occasional multi-chain interactions (i.e. constraint release motion). This indicates that the motion of the cyclic molecule at the whole chain level is distinct from that of the linear chain because of the topological constraint of the cyclic chain under entangled conditions rather than topology-dependent local chain interaction.

We observed diverse conformational states of the cyclic chains because of the topological constraint of the cyclic chains, including open circular and folded linear forms (Fig. 6c). Since the overall motion of the chain is governed by topological constraint-induced modes of chain interactions (i.e. normal entanglement or threading of the cyclic chain by matrix chains), such diverse conformers would result in a large heterogeneity in the overall motion of the chain. This discussion is supported by previous single-molecule studies that reported a broader distribution of diffusion rate of the cyclic molecules in entangled solution of linear chains compared with their linear counterpart[5,21,54,55].

The cyclic chains under entangled conditions sometimes showed larger local displacement of the chains (Fig. 7a, d). We observed a retraction and elongation of the loop-like region of the chain (Fig. 7b, white squares, 7c). This amoeba-like motion has been proposed to explain elastic properties of cyclic chains under entangled conditions such as a cyclic tracer molecule in a melt of linear matrix or in a gel[17,18,56]. Our direct visualisation of the local chain motion demonstrated the existence of this hypothesised but unproven mode of the chain motion. The width of the elongated loop-like region is close to the width of the estimated tube diameter (165 nm, Fig. 7c). The elongated looped chain after the retraction (Fig. 7c, blue line) occupied a space slightly different from the space occupied by the loop initially (Fig. 7c, black line). This mode of motion is predicted for unthreaded cyclic chains under entangled conditions[57]. Together our data demonstrate the existence of the loop-like double-folded linear region in the chain, which is spatially confined in a way similar to linear chains in the reptation model.

We also found that the molecule in open circular and double-folded linear conformation can be converted to each other (Fig. 7d, e). Interestingly, the conversion between the two forms initiated from the end of the double-folded chain (Fig. 7e, arrowheads). This molecule eventually formed two loop-like double-folded regions (Fig. 7e). Both loops are spatially confined to narrow regions with the width close to the tube diameter (127–238 nm, Fig. 7f, see Methods for detailed calculation). The result demonstrates the existence of the multi-arm double-folded conformational state. Although the existence of these conformational states has been postulated[16,56,57], a theoretical framework describing their dynamic behaviours including the conversion between the forms has not been established. A diverse topological interaction involved in entangled cyclic polymer dynamics and the lack of experimental methods to verify theoretical prediction

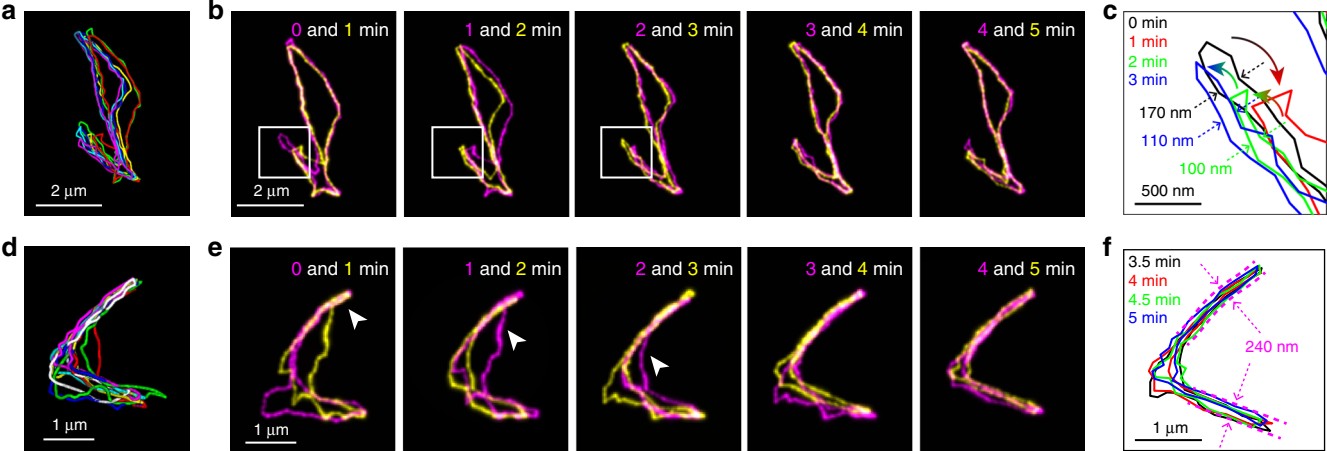

**Fig. 7** Diverse conformation and motion of the entangled cyclic DNA molecules. **a** Superimposed 3D contours of the Cy5-labelled Charomid 9-42 DNA in an aqueous solution containing 7 mg ml$^{-1}$ concentration of non-labelled lambda DNA. **b** 2D projected contours of the DNA molecule displayed in **a**. The local widths of the contours correspond to the standard deviations of the Gaussian at each local segment. Two contours obtained from adjacent time frames are shown in each panel in cyan (earlier time points) and yellow (later time points). An enlarged view of the areas highlighted by the squares is shown in **c**. **c** Enlarged view of the highlighted region in **b**. **d** Superimposed 3D contours of the Cy5-labelled Charomid 9-42 DNA in an aqueous solution containing 5 mg ml$^{-1}$ concentration of non-labelled lambda DNA. **e** 2D projected contours of the DNA molecule displayed in **d**. The local widths of the contours correspond to the standard deviations of the Gaussian at each local segment. Two contours obtained from adjacent time frames are shown in each panel in cyan (earlier time points) and yellow (later time points). **f** 2D projected contours of the DNA molecule displayed in **d** obtained between 3.5 and 5 min. The dashed lines show virtual tubes with a diameter of 260 nm

have been the obstacle to establish the theories describing entangled topological polymer dynamics. Our results demonstrate that the new experimental approach developed in this study provides experimental platform to address key questions in the entangled topological polymer dynamics.

The position-dependent analysis shows that the local motion of the cyclic chains (i.e. amplitude and timescale of the local chain motion in each entangled segment) is independent of the chain position (Fig. 6f, g, see Methods for details of the analysis), which is in contrast to the large position-dependent local motion observed for the linear chains (Fig. 5c, d). The comparison of the linear and cyclic chains clearly highlights the critical contribution of the chain ends to their motion both at length scales larger than the tube diameter and at length scale smaller than the tube diameter.

## Discussion

In this study, we develop new experimental method to characterise entangled polymer dynamics based on time-lapse SR fluorescence localisation microscopy and recently developed single-molecule tracking technique, CA tracking. The time-lapse diffraction-unlimited imaging together with the powerful image processing tools allow us to determine the contours of the entangled chains and their displacement occurring in the time-scale of several minutes with 6–33 nm precision. The unique combination of the SR fluorescence localisation microscopy and CA tracking enables us to characterise the amplitude and time-scale of the local chain motion along the contours with temporal resolution of milliseconds and spatial resolution of tens of nanometres.

Using this new method, we demonstrate that the motion of the linear chains under entangled conditions occurring at the level of both whole chain and entangled segment are consistent with the reptation model. We also show that the entangled linear chains unexpectedly display gradual increase of the local chain motion towards the chain ends, suggesting the gradual change in the confinement level of the chain over tens of entanglements from the chain ends. Since standard polymer physics theories do not

consider such position-dependent chain motion and relaxation, our finding underscores the importance of developing polymer dynamics theories that incorporate the effect of the position-dependent motion.

Our results suggest that the motion of the cyclic molecule at the whole chain level is distinct from that of the linear chain because of the topological constraint of the cyclic chain under entangled conditions rather than topology-dependent local chain motion and interaction. We also prove the existence of hypo-thesised but unproven modes of the motion of cyclic chains under entangled conditions, including amoeba-like motion of double-folded loop-like region. All these results demonstrate that the new experimental platform developed in this study offers unprece-dented opportunities to address key questions in less-well-understood entangle topological polymer dynamics.

Our new experimental platform is, in principle, applicable to a wide range of entangled polymers, including branched polymers, multicyclic polymers, and knotted polymers. We envision that our new approach could be a powerful means to characterised rheological properties of topological polymers under entangled conditions through a direct capture of their nanoscopic and subchain-level motion. The method could also be applicable to address related questions in more complicated system such as polymer gels and cytoskeletal network in cells.

## Methods

**Fluorescent materials**. The 3D calibration of the z-axis positions was done by using TetraSpeck fluorescent microspheres (diameter = 0.1 μm), (excitation/emis-sion peaks—360/430 nm (blue), 505/515 nm (green), 560/580 nm (orange), and 660/680 nm (dark red)) were purchased from ThermoFisher Scientific (T7279). The microspheres were diluted with Milli-Q water to a concentration of $1 \times 10^6$ particles per dm$^3$ then deposited 50 μl of solution on a cleaned coverslip and let dry at room temperature overnight. Lambda DNA (48.5 kbp) (500 ng μl$^{-1}$) was pur-chased from New England Biolabs (N3011S) and Charomid (42 kbp) DNA was purchased from Nippon Gene (Tokyo, Japan). The Topoisomerase-I from New England Biolabs (M0301S) was used to prepare the relaxed form. The DNA molecules were labelled with *Label* IT® Nucleic Acid Labeling Cy5 (MIR 3700) were purchased from Mirus Bio.

**Sample preparations**. To prepare the linear DNA matrix, a 85 μg of lambda DNA (500 ng μl$^{-1}$), 85 μl (1/2 × vol.) of sodium acetate solution (3 M) and 340 μl (2 × vol.)

of isopropanol were added into the reaction mixture and mixed well. The mixture was ultra-centrifuged at 15,000 rpm for 20 min at 4 °C and the supernatant was carefully decanted. The DNA pellets were washed three times with 70% of ethanol and centrifuged at 15,000 rpm for 15 min at 4 °C and the supernatant was carefully decanted. The obtained pellets were dried in air and dissolved in (4.25 µl) TE buffer (10 mM Tris-HCl (pH 8.0) and 0.1 mM EDTA). This gave a final DNA concentration of 17 mg ml$^{-1}$. The solution was incubate at 4 °C overnight.

To prepare the cyclic DNA form from supercoiled form, 10 µg of Charomid 9-42 DNA (42 kbp) were added into 20 µl of the digestion buffer containing 20 units of Topoisomerase-I (New England Biolabs) and 2.5 µl of 10× CutSmart buffer. The reaction mixture was incubated at 37 °C for 15 h. The enzyme was deactivated by incubation at 65 °C for 10 min. The DNA samples were then precipitated using the isopropanol precipitation method, dried in air, and dissolved in 10 µl TE buffer, which gave a final DNA concentration of 1 mg ml$^{-1}$.

Fluorescence labelling of DNA was conducted using a commercially available labelling kit (MIR 3700, Mirus Bio). Prior to use the Label IT Cy5, the vial was warmed to room temperature and the pellet was collected by a quick spin. Then we added 100 µl of reconstitution solution to the pellet in the vial, mixed well, and performed a quick spin. The labelling reaction was prepared by adding 35 µl Milli-Q water, 5 µl 10× Labelling Buffer A, 5 µl of 1 mg ml$^{-1}$ 42 kbp Charomid DNA or 10 µl of 500 ng µl$^{-1}$ 48.5 kbp lambda DNA, and 10 µl of Label Cy5. The labelling density of 5–15 bp/dye molecule was achieved by incubating the reaction at 37 °C for 2 h. The labelled sample was purified by ethanol precipitation method. We brought the final volume to 100 µl by adding 1× Mirus labelling buffer A to the sample. We added 20 µl (0.1 × vol.) of NaCl (5 M) and 400 µl (2 × vol.) of ice-cold 100 % ethanol, mixed well. Then the sample was incubated at −30 °C for 30–45 min, ultra-centrifuged at 15,000 rpm for 30 min at 4 °C, and the supernatant was carefully decanted. Five hundred microlitres of room-temperature 70% ethanol was added and the sample was centrifuged at 4 °C in 15,000 rpm for 15 min. The supernatant was carefully decanted. These steps were repeated three times. The obtained pellets were immediately dissolved in 1000 µl of TN buffer (50 mM Tris (pH 8.0) and 10 mM NaCl) to give a final DNA concentration of 5 µg ml$^{-1}$.

The switching buffer for the SR imaging experiments was prepared immediately before the imaging experiment, which contains an oxygen scavenging system (40 µg ml$^{-1}$ catalase (Sigma-Aldrich), 0.5 mg ml$^{-1}$ glucose oxidase (Sigma-Aldrich), and 10% (w/v) glucose) with a reducing reagent 200 mM β-mercaptoethanol (Sigma-Aldrich) in a TN buffer.

To prepare the DNA sample for 3D SR fluorescence imaging experiment, a 0.5 µl of the fluorescently labelled DNA (5 µg ml$^{-1}$), 0.25 µl of NaCl (0.2 M), solution and 3.5 µl switching buffer were added into the solution of the matrix DNA (4.25 µl containing 10–20 mg ml$^{-1}$ DNA). The solution was gently mixed. The final concentration of the fluorescently labelled DNA mixed with the non-labelled matrix DNA was 5–10 mg ml$^{-1}$. We prepared two different combinations of the labelled DNA and the matrix DNA: linear labelled DNA in the linear matrix DNA (L-L); and cyclic labelled DNA in the linear matrix DNA (C-L). The DNA sample was sandwiched between clean coverslip and glass slide and was sealed by a double-sided adhesive (Grace-Biolabs), which also served as a spacer and providing a sample thickness of 0.12 mm.

SR imaging experiment on a stretched DNA was conducted by depositing fluorescently labelled lambda DNA on a glass coverslip. The glass coverslip were cleaned using ultrasonicator (P60H, Elma Schmidbauer GmbH). The coverslips were sonicated alternatively in absolute ethanol and 1 M KOH solution for 15 min each. We repeated this step two times. We rinsed the cover slips with Milli-Q water seven times after each sonication step. Lastly, the coverslips were sonicated in Milli-Q water for 10 min. After decanting Milli-Q water, 2% 3-aminopropyltriethoxysilane diluted in acetone solution was added to the coverslips, shaken for 30 s, and rinsed with 1 l of Milli-Q water. Lambda DNA (500 ng µl$^{-1}$) was diluted to 100 ng µl$^{-1}$ in Milli-Q water. DNA solution was incubated at 90 °C for 15 min to denature dsDNA into single-stranded DNA (ssDNA). Then, the ssDNA solution was held in a chilled water bath for 2 min to prevent strands from reannealing. Five microlitres of ssDNA solution (blocking DNA 100 ng µl$^{-1}$) was applied at the intersection of a functionalised coverslip and an 18 mm square coverslip that was tilted at 45° to the functionalised coverslip surface. The blocking DNA was dragged across the functionalised coverslip surface by gently moving the 18 mm square coverslip over the functionalised coverslip and allowed to dry if needed. Five microlitres of fluorescently labelled lambda DNA with Cy5 dye (containing 0.2 ng µl$^{-1}$ DNA) was then applied on the functionalised coverslip in the same way. The sample was sandwiched between the functionalised coverslip and glass slide and was sealed by a double-sided adhesive.

**3D SR fluorescence imaging experiment**. The 3D astigmatism-based SR fluorescence localisation imaging experiment was carried out on a custom-built wide-field epi-fluorescence microscope set-up (Fig. 2a)[58]. Two solid-state lasers were used for illumination: a CW 150 mW 638 nm and a CW 150 mW 405 nm (MLD$^{TM}$, Cobolt). In this set-up, the laser beams pass through an acousto-optic tuneable filter (AA Optoelectronic), which allows the intensity of the individual laser lines to be independently controlled using Andor iQ imaging software. The collimated laser beams are introduced into the microscope through an achromatic focusing lens, providing widefield Köhler illumination of the sample. The fluorescence images were recorded using an inverted IX71 microscope (Olympus) with a ×100 numerical aperture 1.49 oil-immersion objective lens. The fluorescence

emitted from the sample was collected again by the same objective lens, sent to the side port of the microscope, and passed through OptoMask (Andor Technology) to enable the capture of crop images using the iXon Ultra 897 EM-CCD camera (Andor Technology). The fluorescence light is separated from the laser excitation light using a dichroic mirror (FF660-Di02-25 × 36) and an emission bandpass filter (FF01-697/58-25) obtained from (Semrock). A cylindrical lens with focal length of 200 mm was placed in front of the iXon Ultra EMCCD camera to enable 3D astigmatism-based SR fluorescence localisation microscopy. The calibration data were recorded at ±500 nm with 10 nm step size using the TetraSpeck fluorescent microspheres sample. The z-axis positions in the acquisition of the calibration data were controlled by a piezo nanopositioning stage (APZ-X00 Piezo Z-Stage) (Andor). The image acquisition was done using the SOLIS imaging software (Andor). The axial drift during data acquisition was controlled by inserting commercial C-Focus lock system (Mad City Labs), mounted directly into the microscopy stage and objective (Supplementary Fig. 1).

Fluorescence images of the fluorescently labelled DNA were captured at low excitation power (<0.1 kW cm$^{-2}$) using the 638 nm laser line as the excitation light. A spatially isolated single fluorescently labelled DNA in the sample was placed at the centre of field of view by using the motorised microscope stage. After capturing a fluorescence image of the sample using the conventional epi-fluorescence microscopy, the laser intensity was increased up to 25–30 kW cm$^{-2}$. Once the density of fluorescence spots reached an appropriate level (i.e. several fluorescent spots in each image) due to the spatiotemporal switching of the fluorescence signal from each Cy5 dye under this illumination condition, we started recording single-molecule fluorescence images of the sample. We used 100 × 50 pixel of the iXon Ultra 897 EM-CCD camera for the imaging experiments. With this size of region of interest, we were able to record and read-out a single image with maximum 0.7 ms speed. The images were recorded at frame rates of 1.0–1.4 kHz. During the image acquisition, in addition to 638 nm laser (25–30 kW cm$^{-2}$) for exciting Cy5 dye, the sample was irradiated by a weak 405 nm laser (0–12 W cm$^{-2}$) to promote switching of the Cy5 dye from its dark state to the bright state. The fluorescence images were recorded with a pixel size of 100 nm and 300 EM gain. A total of 10,000 images were recorded every 1 min to capture the nanoscale motion and relaxation of DNA molecules occurring in the timescale of minute to 10 min. Each SR image was reconstructed from 10,000 raw images, corresponding to a total acquisition time of ~7 s for each reconstructed image. The sample was illuminated only during the image acquisition to reduce the photobleaching of the sample.

**Image analysis**. Once the scatter of localisations in all frames is reconstructed into raw super-resolved images, we traced the exterior (linear and circular DNA molecules) and the interior (circular DNA molecules) boundaries of the DNA molecules by Moore-Neighbor tracing algorithm (bwboundaries; a built-in function in MATLAB, Supplementary Fig. 2)[59]. This step helped to remove extraneous objects and noise that can affect the analysis. To calculate the contour of the DNA molecule, we first divided the molecule into segments of 100 nm length. We then determined the local orientation of each segment by fitting the scatter of localisations to piecewise linear functions; where the fitting is guided by the segments' ends (Supplementary Fig. 3). We, then, fit the scatter in each segment to 1D Gaussian function in perpendicular direction to each slope so that different segments along the molecule can receive different fitting direction, which is based on their local orientations. The calculated centroids obtained after the 1D Gaussian fitting are then connected by linear lines to construct the contour (l-contour) of the DNA molecule (Figs. 3a, b, 4a, 6a, and 7a, d), which corresponds to primitive paths of the DNA molecule.

The amplitude of the local chain motion in each 100 nm length segment was characterised by generating a standard-deviation-weighted contour. The standard deviations (SD$_s$) of the 1D Gaussian fitting representative of each segment (s) were used to generate a standard-deviation-weighed contour. To construct an image that shows a standard-deviation-weighed contour (s-contour), we performed a Gaussian smoothing on the l-contour with a standard deviation of 1 for all SD$_s$ below the mean SD$_s$ ($\mu$-SD$_s$) and a standard deviation of SD$_s$/$\mu$-SD$_s$ for all SD$_s$ above the mean $\mu$-SD$_s$ (Figs. 3e, c, 4d, e, 6b, c, and 7b, e). To calculate the position-dependent standard deviation, we divided the contour into seven regions (i.e. each section in approximately 14% of the contour of individual DNA molecules) and conducted the analysis of the SD$_s$ in each region (Figs. 5c and 6f).

The timescale of the local chain motion in each 100 nm length segment was characterised by characteristic time obtained by the CA tracking. To calculate the characteristic time in each segment, we applied 1D CA tracking method. In principle, the temporal appearance of the localisations across the frames and with respect to each segment carries information on the timescale of local motion of the molecule. Thus, for each segment, we analyse the localisations inside a virtual square whose axis of symmetry is the contour of the segment. The localisations inside each square are then temporally mapped onto 5 nm, 1D virtual square lattice where the CA difference is calculated (Fig. 3f)[21,37]. The timescale for the CA tracking analysis is an average timescale that is equal the timescale of all frame divided by the average number of localisations inside the squares. To calculate the position-dependent characteristic time, we divided the contour of the linear and cyclic DNA into 15 and 10 regions (i.e. each section is approximately 6 and 10% of

the contour of individual linear and cyclic DNA molecules) and conducted the analysis of the characteristic times in each region (Figs. 5d and 6g).

To calculate the position-dependent displacement, we aligned the CM across the consecutive contours. We then calculated displacement between points that lie at equal distances from the aligned CMs. In practice, we draw concentric circles of increasing radii and whose centres are the CM. The intersections of each circle with the contours define the points, which we used to construct the displacement trajectory (Supplementary Fig. 4, Figs. 5a and 6e). To calculate the position-dependent displacement of the aligned 3D contours, we drew concentric spheres of increasing radii and whose centres are the CM. The intersections of each sphere with the contours define the points which we used to construct the displacement trajectory (Fig. 5b).

For the analysis of position-dependent chain motion (i.e. position-dependent displacement of the chain occurring along the direction perpendicular to the contours during the time scale of capturing consecutive SR images, position-dependent standard deviation of the Gaussian, and position-dependent characteristic time obtained from the CA tracking analysis), we used the linear molecules whose contour is longer than 60% of the contour length of the molecule (9.9 μm) to ensure reliable analysis of the position dependence. For the position-dependent chain motion analysis on the cyclic molecules, we used the cyclic molecules whose contour is longer than 60% of the contour length of the molecule (8.6 μm) to ensure reliable analysis of the position dependence, similar to the analysis of the linear chains. Because the starting points of the chains were determined arbitrarily, mean position-dependent data (Fig. 6e–g) do not provide information about the position dependence. We therefore included the position-dependent data obtained from each molecules in these figures. The data obtained from each molecules clearly suggest the absence of position-dependent motion of the cyclic chains, which is in contrast to the large position dependence observed for the linear chains.

**Calculation of the tube diameters.** Lambda DNA has been reported to follow the reptation prediction for concentrations above 0.3–0.63 mg ml$^{-1}$ [32,45]. Since the concentration range of the lambda DNA used in this study is much higher than this threshold (5–10 mg ml$^{-1}$), the tube theory can be applied to estimate the entanglement lengths. Entanglement numbers ($Z$, number of entangled strands in the chain) of 7 and 22 were obtained for lambda DNA at concentrations of 0.8 and 2 mg ml$^{-1}$ in previous studies [44,45]. Based on the reported scaling behaviour [45], we extrapolated the entanglement numbers at the concentrations of 10, 7, and 5 mg ml$^{-1}$ to be 158, 100, and 63. We calculated the entanglement lengths that correspond to the tube diameters ($a$) by the equation, $aZ = <L>$ [3], where $<L>$ denotes primitive path length. Using the maximum (15 μm) and mean (8 μm) lengths of the contours determined experimentally in Fig. 6d as the primitive path lengths, the entanglement lengths and therefore tube diameters are estimated to be 51–95, 80–150, and 127–238 nm for the entangled solutions of lambda DNA at 10, 7, and 5 mg ml$^{-1}$ concentrations, respectively.

**Calculation of the entanglement times.** According to the reptation theory, entanglement times ($\tau_e$, i.e. Rouse time of the entangled strand) is described by the equation, $\tau_e = \tau_r / Z$ [51], where $\tau_r$ denotes Rouse rotational relaxation time. $\tau_r$ of 19, 48, and 184 s were reported for lambda DNA at the concentrations of 1, 1.47, and 2.24 mg ml$^{-1}$ in the previous paper [44]. Based on the reported scaling behaviour [44], we extrapolated $\tau_r$ at the concentrations of 10, 7, and 5 mg ml$^{-1}$ to be 11,900, 4,350, and 683 s. $\tau_e$ is therefore estimated to be 0.47, 0.43, and 0.42 s for lambda DNA at the concentrations of 10, 7, and 5 mg ml$^{-1}$.

## Data availability

The data and the computer codes that support the finding of this study are available from the corresponding author on reasonable request.

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

## Acknowledgements

The research reported in this publication was supported by King Abdullah University of Science and Technology (KAUST) and the KAUST Office of Sponsored Research (OSR) under Award No. OSR-2015-2646-CRG4.

## Author contributions

S.H. conceived the project. M.A. and S.H. designed the experiments. M.A. built the 3D super-resolution imaging set-up, conducted all the experiments, and analysed the data. M.F.S. developed the methods and tools for data analyses and visualisation and wrote the Matlab scripts. M.A., M.F.S., and S.H. analysed the data. S.H. wrote the manuscript with M.A. and M.F.S. All authors discussed the results.

## Additional information

**Competing interests:** The authors declare no competing interests.

