## [Peer Review File · Nature Communications]

Reviewers' comments:

Reviewer #1 (Remarks to the Author):

In this article, the authors combined super-resolution microscopy with cumulative-area tracking for the observation of polymer diffusion. They used linear and circular dsDNA as model systems. They monitored conformational change of single DNA polymer and compared it with reptation theory. The experiment is solid but I have a few concerns. The power of Single-molecule study is to provide heterogeneity of molecules. Thus, it is not a surprising that the authors found unseen conformational heterogeneity. The importance should come from giving the proper explanation on the effect of such heterogeneity or providing a proper theoretical approach to explain this. This study demonstrates only the heterogeneity of polymers. The main claim of this paper is to provide ne experimental approach, which may address key questions in the entangled topological polymer dynamics. In my opinion, to justify its publication in the high standard journal, like Nat. comm., the authors should use their method to answer one of the key questions. But I didn't find a critical findings in this article, which will give more general interest to the community. The method is interesting and looks robust, but combining super-resolution with CA tracking is not a completely noble technique but rather straightforward to me.

1. In page 5. The authors claim that Cy5 labeling has no effect of DNA structure. Dye has a charge. Because the authors labeled cy5 every 5-15 bp, this should have effect one overall movement of DNA. The authors need to justify this statement.
2. In page 8. The authors mentioned that the width of the Gaussian is determined by localization error and local motion of the chain. The authors assumed that the localization error is a fixed value, so the width they contributed to the local motion. However, the localization error varies largely depending on the photon number, which is very much heterogeneous. Thus, the authors need a solid validation on this.
3. In Fig. 4d-f. how many DNA molecules were observed? Did authors include all DNA data or they included only the DNA that had movement? Should be clarified in Fig caption or in text.

Reviewer #2 (Remarks to the Author):

This paper deals with microscopic polymer dynamics at the single chain level at a maximum spatial and time resolutions of 6-30 nm and 0.7 ms, respectively. The authors report reptation motion for linear and cyclic DNA chains in 3D and extracted this 3D motion from 2D projected images. Their analysis is based on a nowadays standard super-resolution microscopy method with an extra cumulative area tracking, covering the whole range of length and time scales in the reptation motion. They revisited the reptation motion of the linear chains in a semi-diluted solution where they observed a chain-position-dependent motion of the entangled linear chain beyond the reptation model. Moreover, they demonstrate the motion of cyclic chains in entangled environments.

The experimental and analysis approach put forward here is timely and paves the way towards extracting information in the third dimension with a high temporal resolution. I believe this paper is a valuable piece of work that can be even further improved by 1) a more in-depth comparison between the results obtained on the linear and cyclic chains and 2) by comparing even more to theory. There are a handful of studies (mostly theoretical) available that address the constraint release in polymer reptation in the physical networks in which all neighbouring chains are mobile. Such comparison between the parameters extracted via experiment and theoretical models can strongly enrich this study and make a platform for further studies. Summarizing, if the authors can address my comments

below and after adding the appropriate references, I enthusiastically recommend publication of this manuscript.

Detailed comments:

1. How did the authors control the focus using the piezo nano-positioning stage, was it programmed or manually? If programmed is that considered in the analysis, does it have any effect on the results?
2. How is the stability of the DNA chains in time? What is the frequency of the breaking of the chains and how would that affect the entanglement length?
3. There have been several characteristic time scales defined for the reptation motion. I advise the authors to define them and make a comparison between the time scales achieved here to those reported in literature. Are they in agreement with the experimental values reported so far?
4. The same is for the confining potential or the tube diameter (for instance see e.g. refs: experimental: Confining Potential as a Function of Polymer Stiffness and Concentration in Entangled Polymer Solutions, *J. Phys. Chem. B*, 2017, 121 (22), pp 5613–5620; and Simulation: Quantifying chain reptation in entangled polymer melts: Topological and dynamical mapping of atomistic simulation results onto the tube model, *The Journal of Chemical Physics* 132, 124904 (2010))
5. The primitive path has to be described in the data treatment, what has been attributed to primitive path, is it the spline going through the Gaussians? Is there such a concept existing for the cyclic chains as well?
6. On page 10, instead of “under the condition of our experiment” the condition of observing such confined motion has to be described e.g. that is when the molecular weight of the diffusing entity is larger than the entanglement molecular weight.
7. On page 10 where the authors mention “Occasionally, we observed small chain regions that exhibited the displacement larger than the tube diameter” This is most likely related to the constraint release mechanism. In an entangled solution topological constraints are formed by the reptating neighboring chains. Therefore, some of the chains constructing the tube reptate away while the tracer chain is reptating within the tube. This change in the topological environment induced by the mobility of the surrounding chains permits the tracer chain an extra mobility. This mechanism is currently under debate and it would be very interesting if the authors could use different models for defining the diffusion constants at different time scales and see which fits experimental results best (for details see Reptation and Constraint Release in Linear Polymer Melts. An Experimental Study, J. von Seggern S. Klotz and H.-J. Cantow).
8. The larger motional freedom at the end of the chains has been observed in several studies and is attributed to the larger degree of freedom at both ends. References need to be added here.
9. It would be beneficial to the reader if the authors could project their 3D data in x-z/y-z plane and compare the results to the xy analysis to see how the information extracted from the third dimension is affecting the results. This is one of the key aspects in this paper and I strongly recommend addressing the benefits by having access to the third dimension. This can be done in supporting info or in the main text.
10. How do the authors define the contour length of the cyclic chains as I suppose the nearest neighbor method would not work here?
11. The effect of the position dependent motion can be considered as the temporal heterogeneity or the rearrangement of the entanglements as well. In spite of their occurrence in such small time scales relative to reptation time, it seems that they are playing an important role in polymer dynamics. Moreover the free ends of the chains shall be considered as an important effect in polymer motion as their dynamics can be influenced/controlled by introducing heavier groups or branches to the ends. That could also be tested via this analysis.
12. In figure 2d it would be good to place the coordinates of the x-y axis as guide to the eye, it is easier to follow the motion of the chain in that way. In figure 4a please indicate the times at which each snapshot has been taken.

Reviewer #3 (Remarks to the Author):

The authors aim to investigate the entanglement of polymers and related theories by developing a single-molecule super-resolution fluorescence assay for measuring the configurations of individual dsDNA molecules. The authors introduce techniques allowing for 3D measurements of 10,000 frames and claim a precision in measuring polymer chain contours in the order of 6-33 nm. A key motivation of the work is to provide experimental techniques to move beyond ensemble measurements (like light/neutron scattering, NMR, or viscosity/stress-relaxation responses) in order to collect more refined measurement data that explores individual polymer chain motions within entangled systems. The authors aim to use this data to make comparisons quantitatively and qualitatively with hypothesized entanglement mechanisms and polymer theories. They find qualitative evidence supporting a hypothesized mode for cyclic chains under entangled conditions that deviates from traditional reptation theory.

While overall the paper is well-written and the techniques of potential interest, a lot of the discussions and analysis are not developed in detail. The authors also do not sufficiently survey the related literature of prior works in related areas, and explain how their approaches improve, differ, or are the same. In the literature on single-molecule chain fluctuations there has already been extensive work not only on tracking chain contours but also on extracting additional information, such as mechanical properties including the persistence length for microtubules and actin filaments. The authors also do not include much discussion of this extensive prior work either in the Introduction or in the other sections discussing their techniques. For instance, the authors should see the following relevant papers, among others:

"B Mickey, J Howard, Aug 1995, "Rigidity of microtubules is increased by stabilizing agents.", *The Journal of Cell Biology*, 130 (4) 909-917."

"D. Valdman, B.J. Lopez, M.T. Valentine, P.J. Atzberger, Force spectroscopy of complex biopolymers with heterogeneous elasticity, *Soft Matter* 9 (2013) 772–778."

"H. Hsu, P. Wolfgang and K. Binder, Breakdown of the Kratky–Porod wormlike chain model for semi-flexible polymers in two dimensions, *Europhys. Lett.*, 2011, 95, 68004."

"D. Valdman, P. J. Atzberger, D. Yu, S. Kuei and M. T. Valentine, Spectral analysis methods for the robust measurement of the flexural rigidity of biopolymers, *Biophys. J.*, 2012, 102, 1144–1153."

"F. Gittes, B. Mickey, J. Nettleton and J. Howard, Flexural rigidity of microtubules and actin Filaments measured from thermal Fluctuations in shape, *J. Cell Biol.*, 1996, 120, 923–934."

"Brangwynne, C. P., G. H. Koenderink, D. A. Weitz. 2007. Bending dynamics of fluctuating biopolymers probed by automated high-resolution filament tracking. *Biophys. J.* 93:346–359."

The authors also appear to use an analysis method for estimating the polymer chain contours that fits locally a Gaussian to cross-sections of the chain using the maximum to construct a contour by connecting linear links to cross-sectional peaks. It is unclear, especially in the 3D setting, how precisely the reference cross-sections were determined a priori and how robust this procedure was to noise. Some of the polymer signals can be seen at the level of resolution of the fluorescence to exhibit

sporadic kinks or other bulging features. For instance in Supplementary Note Figure 2, there is a kink about half-way into the contour. It is unclear that is really indicative of the relevant shape of the DNA strand, some self-interactions/clustering or some other artifact of the measurements. The authors main way to characterize the chain fluctuations was through a technique they called "cumulative-area tracking." (CA) (related to area of the florescent signal swept out by a contour within standard deviations of the contour). Given the observed responses of the polymer, this appears to provide primarily a qualitative characterization of the polymer chain configurations. While the CA could be useful to gain some qualitative insights into the chain configurations, it was unclear why the authors did not develop more refined analysis methods for their sophisticated assay. It would seem the assay is capable in principle of producing much more detailed quantitative data than just the CA that was reported which could be compared to polymer theories.

Overall, the paper does appear to report some interesting novel observations of fluctuations of polymer chains subject to entanglements. However, it seems much more could have been done in the development of the techniques for analysis and processing of the measurements to provide more quantitatively accurate data sets. Overall, the paper is well-written and the techniques and findings appear of potential interest.

Point-by-point response to the referees

Reviewer 1

1. *The experiment is solid.*

Thank you for the positive comment on our work.

2. *The power of Single-molecule study is to provide heterogeneity of molecules. Thus, it is not a surprising that the authors found unseen conformational heterogeneity. The importance should come from giving the proper explanation on the effect of such heterogeneity or providing a proper theoretical approach to explain this. This study demonstrates only the heterogeneity of polymers. The main claim of this paper is to provide ne experimental approach, which may address key questions in the entangled topological polymer dynamics. In my opinion, to justify its publication in the high standard journal, like Nat. comm., the authors should use their method to answer one of the key questions. But I didn't find a critical findings in this article, which will give more general interest to the community.*

The primary aim of this study is to develop new technique for the capture and characterization of nanoscopic polymer dynamics under entangled conditions and apply the method to address important questions in polymer physics. As the reviewer mentioned, we reported conformational heterogeneity of the cyclic polymer chains under entangled conditions. Although this is one of the important findings in this study, we also reported other important new findings in this paper. This includes chain-position-dependent relaxation dynamics of linear chains under entangled conditions, constraint-release-type motion of the chains, topological-constraint-induced lateral motion of cyclic chains. The position-dependent relaxation of polymer chains under entangled conditions is beyond the standard polymer physics theory (i.e. reptation theory), which implies a key contribution of mutual chain relaxation in the entangled polymer dynamics. The constraint-release motion is a controversial issue in the relevant filed, and our finding strongly suggests that this mechanism is operative in the entangled polymer dynamics. The origin of topology-dependent viscoelastic properties of polymers has been an important issue in the relevant field, and our result indicates that topological constraint plays a major role in this. Together, we reported important new scientific insights about entangled polymer dynamics in this paper in addition to the development of the new method. In the revised manuscript, we modified the main text and emphasized these new scientific findings so that readers will be able to understand our new findings easily.

3. *The method is interesting and looks robust, but combining super-resolution with CA tracking is not a completely noble technique but rather straightforward to me.*

The method we developed in this study (i.e. super-resolution fluorescence imaging combined with cumulative-area tracking) could be relatively straightforward. However, to the best of our knowledge, we utilized for the first time the temporal dynamics of the localization in the super-resolution imaging as a means to characterize nanoscopic local motion of polymer molecules. This enabled us to quantify the motion and relaxation of individual entangled polymer chains occurring at a wide range of time and length scales in real space, which is prerequisite for the analysis of complicated polymer dynamics occurring under entangled conditions.

4. *The authors claim that Cy5 labeling has no effect of DNA structure. Dye has a charge. Because the authors labeled cy5 every 5-15 bp, this should have effect one overall movement of DNA. The authors need to justify this statement.*

Thank you for the comment. It is known that charges on DNA influence its motion and relaxation in a solution, in particular under entangled conditions. Given the higher charge density of DNA itself (a negative charge per nucleotide, i.e. two negative charges per bp) compared with that arises from the Cy5 dyes (a positive charge per 5-15 bp), the negative charge of DNA has a major effect on its motion rather than the positive charge of Cy5 dyes. We carefully adjusted the salt concentration in our experiment in such a way that the charge interaction (repulsion) between the DNA chains is effectively screened [Manning GS, Q Rev. Biophys., 1987, 11, 179.; Perkins TT et al., Science, 1994, 264, 819.; Smith DE et al., Phys. Rev. Lett., 1995, 75, 4146.; Teixeira RE et al., Macromolecules, 2007, 40, 2461.; Robertson RM et al., Macromolecules, 2007, 40, 3373.] in order to characterize the motion of the DNA molecules under entangled conditions without a significant effect of the charge interaction. In addition, the Cy5 dyes are conjugated to DNA through a long and flexible linker. Furthermore, the Cy5 dyes intercalate neither into major nor into minor groove of DNA. Thus, elastic properties of DNA are not influenced significantly by the Cy5 labelling. We modified this sentence in the revised manuscript. “Unlike conventional labelling with intercalator dyes, Cy5 dyes were covalently-attached to heteroatoms on DNA molecule through a flexible linker. This allowed fluorescence labelling of DNA with minimum effect on its structure and motion under entangled conditions”.

5. *The authors mentioned that the width of the Gaussian is determined by localization error and local motion of the chain. The authors assumed that the localization error is a fixed value, so the width they contributed to the local motion. However, the localization error varies largely depending on the photon number, which is very much heterogeneous. Thus, the authors need a solid validation on this.*

Thank you for the comment. As the reviewer pointed, the localization error is dependent on the photon number in each localized spot, and therefore, the width of the Gaussian shows a broad distribution (Figure 3c). Since the photon number in each localized spot is determined stochastically (i.e. photoswitching of

the Alexa Fluor 647 dye from the bright to the dark state occurs stochastically), the width of the Gaussian determined by the photon number should be independent on the positions along the contour of the polymer chain when the mean values of the Gaussian width at each position along the chain are calculated using the data obtained from a large number of molecules. If the position-dependent local motion along the chain exists, we will observe position-dependent Gaussian width when mean values are calculated using the data obtained from a large number of molecules, which is shown in Figure 4e. To avoid any confusion, we rephrased this sentence in the revised manuscript. “Since the former contribution is determined by the number of photon detected in each localized spot that is independent on the positions along the contour of the chain, the latter determines the position-dependent width of the Gaussian when mean widths of the Gaussian at each position are calculated from a large number of molecules”.

6. *In Fig. 4d-f. how many DNA molecules were observed? Did authors include all DNA data or they included only the DNA that had movement? Should be clarified in Fig caption or in text.*

Thank you for the comment. We constructed the data displayed in Figure 4d, 4e, and 4f using 9, 15, and 12 independent images of the DNA molecules. We included this information in the caption for Figure 4 in the revised manuscript.

Reviewer 2

1. *if the authors can address my comments below and after adding the appropriate references, I enthusiastically recommend publication of this manuscript.*

Thank you for the positive comment on our work.

2. *How did the authors control the focus using the piezo nano-positioning stage, was it programmed or manually? If programmed is that considered in the analysis, does it have any effect on the results?*

We used a commercially available automatic focus correction system, C-Focus system from Mad City Labs Inc. The system measures the distance between the objective and the bottom surface of the cover slip using an optical sensor attached to the objective and makes the adjustment of the Z-position of the objective lens using the objective piezo positioner. The piezo positioner is positioned through a closed loop control. We confirmed that the system can minimize the stage drift along the z-axis as small as ± 20 nm during 15 min image acquisition (Supplementary Figure 3). Since the stage drift in the length scale of ± 20 nm does not affect the analysis of the data, we did not apply any further image processing before we reconstructed 3D

super-resolution images.

3. *How is the stability of the DNA chains in time? What is the frequency of the breaking of the chains and how would that affect the entanglement length?*

We occasionally observed a cleavage of labelled DNA. The infrequent photocleavage events are due to a low concentration of oxygen in the imaging buffer as the buffer contains an oxygen scavenging system (i.e. glucose oxidase, catalase, glucose). Most of the cleavage events occur during the initial phase of the image acquisition (i.e. during the photoswitching of Cy5 dyes to its dark state before we start capturing single-molecule fluorescence images to reconstruct super-resolution images) because a large number of Cy5 dyes absorb the excitation light in this phase that results in highest probability of photo-induced cleavage reaction of DNA. We note that the cleave of DNA is obvious during this stage as we can see the fluorescence image of the entire chain. Compared with this initial stage of the imaging experiment, the cleavage reaction occurs at much less efficiency at the later stage of the data acquisition (i.e. during the acquisition of single-molecule fluorescence images of Cy5 dyes after reaching the condition for recording the images for super-resolution localization microscopy) because the number of Cy5 molecules that absorb the excitation light is much less than that in the initial phase of the imaging experiment. Therefore, we were able to record 10,000 frames of the fluorescence images without having a problem of photo-induced cleavage of DNA. We also note that the density of the Cy5-labelled DNA is much less than that of non-labelled DNA. Thus, the infrequent photocleavage reaction occurring to the labelled DNA has negligible effect on the entangled motion of DNA.

4. *There have been several characteristic time scales defined for the reptation motion. I advise the authors to define them and make a comparison between the time scales achieved here to those reported in literature. Are they in agreement with the experimental values reported so far?*

Thank you for the comment. As the reviewer pointed, we did not explain in detail the characteristic times that define reptation motion in the original manuscript. According to the reviewer's comment, we added the definition of these key time scales in the revised manuscript as follow, "Reptation motion is defined by several characteristic time scales, including entanglement time (i.e. Rouse time of the entangled strand, τ_e), Rouse rotational relaxation time (i.e. characteristic time during which a polymer molecule diffuses over a distance of its size, τ_r), and disentanglement time (i.e. characteristic time during which a polymer chain moves out of its original tube, τ_d)". We also provided additional discussion on the time scale of the position-dependent chain motion observed in this study by comparing the observed time scale with those related to reptation motion as follow, "A large motional freedom at the chain ends has been captured directly in previous studies using synthetic polymers [Keshavarz M et al. ACS Nano, 2016, 10, 1434.; Keshavarz M et al. J. Phys. Chem. B, 2017, 121, 5613.]. In these studies, the large motional freedom at the chain ends

was observed in the time scale of reptation motion (i.e. $\tau_r < t < \tau_d$). In this study, we observed the chain-position dependent motion at the time scale of minutes, which is much shorter than the estimated Rouse rotational relaxation time (τ_r) of lambda DNA at 7 – 10 mg ml⁻¹ concentration (1.2 – 3.3 hours), suggesting that the chain position-dependent motion observed in this study is not directly related to the reptation motion of the chain”. We note that most of the characteristic times that define reptation motion have not been reported for entangled DNA. Only τ_r has been estimated so far. In most of the previous studies, the reptation motion of entangled DNA was demonstrated by the scaling law of self-diffusion coefficient (i.e. molecular weight and concentration dependent self-diffusion coefficient), which occurs at time scale larger than τ_d . This is probably due to the short Kuhn length of dsDNA (approximately 100 nm), which makes it difficult to experimentally capture the motion of DNA at the time and length scales that are relevant to reptation motion, highlighting the power of our new method to address key questions in entangled polymer dynamics.

5. *The same is for the confining potential or the tube diameter (for instance see e.g. refs: experimental: Confining Potential as a Function of Polymer Stiffness and Concentration in Entangled Polymer Solutions, J. Phys. Chem. B, 2017, 121 (22), pp 5613–5620; and Simulation: Quantifying chain reptation in entangled polymer melts: Topological and dynamical mapping of atomistic simulation results onto the tube model, The Journal of Chemical Physics 132, 124904 (2010))*

Thank you for the comment. As the reviewer pointed, we did not provide enough information about the discussion on the tube diameter in the original manuscript. According to the reviewer’s comment, we revised the manuscript related to the tube diameter as follow, “Reptation motion is characterized by a confined motion of polymer chains along virtual tube, in which the tube diameter is one of the key parameters that characterizes reptation motion [ref. 42, 43]. Given the concentration of the semi-dilute solution (10 mg ml⁻¹) and the scaling law of reptation model [ref. 44, 45], the tracer linear chain has 158 entanglements with the matrix chains, which correspond to the entanglement length and the tube diameter of approximately 51 – 95 nm (see Supplementary Note 1 for detailed calculation). The displacement of the chain between the consecutive time frames along the direction perpendicular to the contour at the centre region of the chain was less than the diameter of the tube, consistent with the prediction by the reptation theory”. We also cited two reference articles (ref. 42 and 43, Keshavarz M et al. J. Phys. Chem. B, 2017, 121, 5613.; Stephanou PS et al. J. Chem. Phys., 2010, 132, 124904.) that the reviewer kindly mentioned in the review report during this discussion. We note that there is no previous study that reported or estimated tube diameter of dsDNA under entangled conditions. Therefore, we calculated the tube diameter based on the entanglement number estimated previously and the primitive path length estimated in this study. The detailed calculation of the tube diameter is provided in the Supplementary Information.

6. *The primitive path has to be described in the data treatment, what has been attributed to primitive path, is*

it the spline going through the Gaussians? Is there such a concept existing for the cyclic chains as well?

Thank you for the comment. The primitive path of the DNA molecules was obtained by connecting the local centroids by linear lines that were obtained by fitting each segment to 1D Gaussian function. Since this point was not very clear in the original manuscript, we mentioned this procedure clearly in the revised manuscript. “The calculated centroids obtained after the 1D Gaussian fitting are then connected by linear lines to construct the contour (l-contour) of the DNA molecule, which corresponds to primitive paths of the DNA molecule.” Given the importance of the detailed image analysis pointed by the reviewer, we moved the description about the data analysis from the Supplementary Information to the main text. The concept of primitive path has been applied to entangled cyclic polymers in a way similar to linear polymers in some previous studies (Yang YB et al. J. Chem. Phys., 2010, 133, 064901.; Halverson JD et al. J. Chem. Phys., 2011, 134, 204905.; Halverson JD et al. Phys. Rev. Lett., 2012, 108, 038301.). We cited these reference papers in the revised manuscript.

7. *On page 10, instead of “under the condition of our experiment” the condition of observing such confined motion has to be described e.g. that is when the molecular weight of the diffusing entity is larger than the entanglement molecular weight.*

Thank you for the comment. According to the suggestion of the reviewer, we modified this sentence in the revised manuscript. “This result suggests that while we captured primitive paths of the chains rather than actual contours of the chains, the primitive-path length of the chains approaches to their contour length due to the high level of spatial confinement of the chains at the concentrations above the entanglement concentration (C_e) of lambda DNA ($C_e \approx 0.6 \text{ mg ml}^{-1}$) [Smith DE et al. Phys. Rev. Lett., 1995, 75, 4146].” We note that the entanglement molecular weight of dsDNA has not been determined. According to Reference 32 [Smith DE et al. Phys. Rev. Lett., 1995, 75, 4146], the entanglement molecular weight of dsDNA has been estimated to be between 9.4 kbp and 23.1 kbp, which is much shorter than lambda DNA (48.5 kbp).

8. *On page 10 where the authors mention “Occasionally, we observed small chain regions that exhibited the displacement larger than the tube diameter” This is most likely related to the constraint release mechanism. In an entangled solution topological constraints are formed by the reptating neighboring chains. Therefore, some of the chains constructing the tube reptate away while the tracer chain is reptating within the tube. This change in the topological environment induced by the mobility of the surrounding chains permits the tracer chain an extra mobility. This mechanism is currently under debate and it would be very interesting if the authors could use different models for defining the diffusion constants at different time scales and see which fits experimental results best (for details see Reptation and Constraint Release in Linear Polymer Melts. An Experimental Study, J. von Seggern S. Klotz and H.-J. Cantow).*

Thank you for the comment. As the reviewer pointed, we believe that the local chain motion whose displacement is larger than the tube diameter is attributed to the constraint release motion. While we mentioned this in the original manuscript (“All these findings are consistent with the reptation model with a small contribution of multi-chain interactions such as constrain release motion”), according to the suggestion of the reviewer, we further extended our discussion on this issue in the revised manuscript as follow. “This observation indicates that constraint release motion, in which the constraints of the chain due to the entanglement is released through motions of the surrounding chains, is operative. According to the previous study on entangled DNA [Smith DE et al. Phys. Rev. Lett., 1995, 75, 4146], the entanglement molecular weight (M_e) of DNA is estimated to be between 9.4 kbp and 23.1 kbp. The molecular weight of lambda DNA (48.5 kbp) is therefore 2.1 – 5.1 times larger than M_e . Previous studies on synthetic polymers in melts suggested that the constraint release motion influences self-diffusion of entangled polymer when the molecular weight of the polymer (M_w) is in the range between $M_e < M_w < 10M_e$ [von Seggern J et al. Macromolecules, 1991, 24, 3300.; Green PF et al. Macromolecules, 1986, 19, 1108.; Smith BA et al, macromolecules, 1985, 18, 1901.]. Therefore, the local chain displacements larger than the tube diameter observed in this study can be interpreted reasonably by the constraint release motion of the chain. To the best of our knowledge, we captured for the first time unambiguous sign of constraint release motion of polymer chains. At the moment, we cannot distinguish distinct constraint release models proposed previously [Graessley WW. Adv. Polym. Sci., 1982, 48, 67.; Klein J. Macromolecules, 1986, 19, 105.; Hess W. Macromolecules, 1986, 19, 1395.] based on our observation. Nevertheless, these results strongly suggest that our imaging platform enables unprecedented subchain level characterisation of entangled polymer dynamics in real space.” We added this discussion in the revised manuscript.

9. *The larger motional freedom at the end of the chains has been observed in several studies and is attributed to the larger degree of freedom at both ends. References need to be added here.*

As the reviewer pointed, large motional freedom at the chain ends has been captured directly in previous studies (for example Keshavarz M et al. ACS Nano, 2016, 10, 1434.; Keshavarz M et al. J. Phys. Chem. B, 2017, 121, 5613.). In these studies, the large motional freedom at the chain ends was observed in the time scale of reptation motion. In the current study, we observed the chain position-dependent motion at the time scale (subsecond to minute) much shorter than Rouse time (in the time scale of hours). This indicates that the chain position-dependent motion observed in this study is not directly related to the reptation motion of the chain, which is distinct from the large motional freedom at the chain ends reported in the previous studies that can be interpreted within the framework of the reptation theory. We added this discussion and cited these previous studies in the revised manuscript.

10. *It would be beneficial to the reader if the authors could project their 3D data in x-z/y-z plane and compare*

the results to the xy analysis to see how the information extracted from the third dimension is affecting the results. This is one of the key aspects in this paper and I strongly recommend addressing the benefits by having access to the third dimension. This can be done in supporting info or in the main text.

Thank you for the comment. While we captured 3D conformation of single chains in our 3D super-resolution (SR) imaging experiment, we selected molecules that have an extended conformation along the XY plane (for example the image shown in Figure 3a). The reason is that we have better spatial resolution along the XY plane compared with XZ and YZ plane in our 3D SR imaging experiment. Thus, by analyzing the motion of individual polymer molecules in the XY plane, we can characterize their motion in a most accurate way. We also note that the maximum Z-axis range covered by our 3D SR imaging is approximately 1 μm , which imposes some restrictions on the use of XZ and YZ data. At the request of the reviewer, we conducted the analysis on the chain displacement (Figure 4d) using the 3D data instead of 2D projected data on the XY plane. The 2D (Figure 4d) and 3D (below figure) data show similar trend. Most importantly, the chain-position-dependent displacement that we observed in the 2D analysis is also seen in the 3D data. This confirms the validity of our analysis using the XY data. We added the 3D data in the Supplementary Information. We also added the description about the procedure of the 3D analysis to the Methods.

Figure. 3D analysis of the position-dependent displacement of the linear DNA molecules under entangled conditions.

11. *How do the authors define the contour length of the cyclic chains as I suppose the nearest neighbor method would not work here?*

We used the nearest neighbour method to set a boundary between the interior (i.e. the region containing a DNA molecule) and the exterior (i.e. the region that does not contain a DNA molecule). Then, the contours of the molecule including the cyclic molecules were determined using a two-step image analysis described

in the Methods (i.e. piecewise linear fitting followed by 1D Gaussian fitting). We first reconstructed super-resolution (SR) images of the cyclic polymer molecules. In most cases, we can see contours of the cyclic chains by visual inspection without any further image processing since the SR images are reconstructed with 30 – 35 nm image resolution. As shown in the figure inserted below, we can easily distinguish two strands in a single cyclic polymer chain that are separated by approximately 30 nm distance by reconstructing the SR images. In these cases, we simply connected the Gaussian peaks along the chains to determine the contours of the cyclic chains. In rare cases, we observed complete spatial overlap of two strands in single cyclic chains (i.e. separation distance between the two strands below 30 nm). We excluded these rare cases from our analysis.

Figure. Cross-section of the SR image obtained from a cyclic DNA molecule. The image shows that two strands in the single chain that are separated by approximately 30 nm distance can be easily distinguished.

12. *The effect of the position dependent motion can be considered as the temporal heterogeneity or the rearrangement of the entanglements as well. In spite of their occurrence in such small time scales relative to reptation time, it seems that they are playing an important role in polymer dynamics. Moreover the free ends of the chains shall be considered as an important effect in polymer motion as their dynamics can be influenced/controlled by introducing heavier groups or branches to the ends. That could also be tested via this analysis.*

Thank you for the inspiring comment. As the reviewer mentioned, our new experimental platform is, in principle, applicable to a wide range of entangled polymers, including a linear polymer with bulky chain ends. The method could also be applicable to polymer chains with other topological states, including branched polymers, multicyclic polymers, and knotted polymers. We envision that our new approach could in fact be a powerful means to characterize rheological properties of topological polymers under entangled conditions through a direct capture of their nanoscopic and subchain level motion. Since these studies are

beyond the scope of our current study, we added these future outlooks to the concluding remark of the paper in the revised manuscript.

13. *In figure 2d it would be good to place the coordinates of the x-y axis as guide to the eye, it is easier to follow the motion of the chain in that way. In figure 4a please indicate the times at which each snapshot has been taken.*

Thank you for the comments. According to the reviewer's suggestion, we added the x-y coordinates in Figure 2d and the time points in Figure 4a.

Reviewer 3

1. *The authors also do not sufficiently survey the related literature of prior works in related areas, and explain how their approaches improve, differ, or are the same. In the literature on single-molecule chain fluctuations there has already been extensive work not only on tracking chain contours but also on extracting additional information, such as mechanical properties including the persistence length for microtubules and actin filaments. The authors also do not include much discussion of this extensive prior work either in the Introduction or in the other sections discussing their techniques.*

Thank you for the comment. We also thank the reviewer to introduce us the previous studies. The main focus of this study is nanoscopic characterization of entangled polymer dynamics rather than conformational fluctuation of polymer chains under dilute conditions (i.e. no entanglement between the chains), and therefore, we focused our attention only to polymer dynamics occurring under entangled conditions in the original manuscript. As pointed by the reviewer, previous studies on the rigidity of biopolymers (actin and microtubule) are technically relevant to our new method, we broadened the scope of this study and introduced these previous works in the revised manuscript. We also note that the persistence length of dsDNA (approximately 50 nm) has already been estimated accurately using various single-molecule methods, including optical tweezers and flow-stretching techniques. In fact, we chose DNA as a model polymer in this study since its elastic properties have been very well characterized.

2. *The authors also appear to use an analysis method for estimating the polymer chain contours that fits locally a Gaussian to cross-sections of the chain using the maximum to construct a contour by connecting linear links to cross-sectional peaks. It is unclear, especially in the 3D setting, how precisely the reference cross-sections were determined a priori and how robust this procedure was to noise. Some of the polymer signals can be seen at the level of resolution of the fluorescence to exhibit sporadic kinks or other bulging*

features. For instance in Supplementary Note Figure 2, there is a kink about half-way into the contour. It is unclear that is really indicative of the relevant shape of the DNA strand, some self-interactions/clustering or some other artifact of the measurements.

Thank you for the comment. As pointed by the reviewer, analyses of fluorescence images including the estimation of the polymer chain contours are usually affected by noise. Unlike most of image analyses that are affected by noise in raw images, our analysis is based on the localized spots in the reconstructed fluorescence images. The accuracy of the analysis is determined by the precision of the localization of each spot (that is determined by number of photon detected from single fluorophores and background noise) and the spatial density of the localized spots. Therefore, the reconstructed super-resolution images (for example Figure 3b) already include the effect of noise. That means the image resolution can be defined by the full width at half maximum of the Gaussian used for the fitting of the reconstructed images. In our case, we obtained 33 nm image resolution. Since the contour of the chain can be determined by connecting the peaks of the Gaussian, the peak determination error of the Gaussian (6 nm), in principle, corresponds to the precision of the determination of the contour. Given the size of each segment for the Gaussian analysis (100 nm), we determined the contours of the chains with 6 – 33 nm precision. Since persistence length of dsDNA is approximately 50 nm (Kuhn length of approximately 100 nm), we expect to observe bending of the chain at this length scale. We note that the kinks that we observed in some images (for example in Figure 3b) are in this length scale. We also note that the line shown in Supplementary Note Figure 2 in the original manuscript (Supplementary Figure 6 in the revised manuscript) is a contour of the molecule roughly estimated by the piecewise linear mapping. The final contour (Figure 3a, 3b) was determined by the Gaussian fitting based on this estimated contour as described in the Methods.

- 3. The authors main way to characterize the chain fluctuations was through a technique they called "cumulative-area tracking." (CA) (related to area of the florescent signal swept out by a contour within standard deviations of the contour). Given the observed responses of the polymer, this appears to provide primarily a qualitative characterization of the polymer chain configurations. While the CA could be useful to gain some qualitative insights into the chain configurations, it was unclear why the authors did not develop more refined analysis methods for their sophisticated assay. It would seem the assay is capable in principle of producing much more detailed quantitative data than just the CA that was reported which could be compared to polymer theories. Overall, the paper does appear to report some interesting novel observations of fluctuations of polymer chains subject to entanglements. However, it seems much more could have been done in the development of the techniques for analysis and processing of the measurements to provide more quantitatively accurate data sets.*

As we mentioned in the abstract of the paper, a new method that enables to capture and quantify the motion and relaxation of individual entangled polymer chains occurring at a wide range of length and time scales

is needed to address key questions in the relatively poorly understood entangled polymer dynamics. By combining super-resolution fluorescence localization microscopy and recently developed single-molecule tracking method in our lab, cumulative-area tracking, we achieved unprecedented range of spatiotemporal resolution, from nanometres/milliseconds to micrometres/minutes. The successful development of this method enabled us to quantify several key parameters in entangled polymer dynamics, including the length and time scale of whole chain motion (for example Figure 4d) as well as local chain motion (for example Figure 4e and f). These quantitative information cannot be obtained by any existing single-molecule techniques. We believe that these examples highlight the power of our new method.

4. *Overall, the paper is well-written and the techniques and findings appear of potential interest.*

Thank you for the positive comment on our work.

Reviewers' comments:

Reviewer #1 (Remarks to the Author):

The authors have improved the paper well in the revised manuscript.

1. about the Cy5 labeling, I agree that its labeling will have a minimal effect on the DNA structure. However, movement is a different story. The authors didn't provide any experimental evidence that the movement of bare DNA is the same or similar with the many Cy5 dyes' labeled DNA (not a single Cy5-labeled DNA). Can the authors show at least their diffusion time in solution is the same using FCS? Or a salt dependent measurement? Otherwise, the comment "minimum effect on its structure and motion under entangled conditions" is not valid yet.

2. In Fig. 4, they used 9, 15, and 12 molecules for the analysis. I am not sure whether this small number is good enough to give a such a solid statistics. Can the authors provide a statistical justification on their analysis using such a small number of sampling?

Reviewer #2 (Remarks to the Author):

I am happy with the answers to my comments/questions. I was already in favor of publishing this manuscript, I am even more enthusiastic now.

Point-by-point response to the referees

Reviewer 1

1. *The authors have improved the paper well in the revised manuscript.*

Thank you for the positive comment on our work.

2. *about the Cy5 labeling, I agree that its labeling will have a minimal effect on the DNA structure. However, movement is a different story. The authors didn't provide any experimental evidence that the movement of bare DNA is the same or similar with the many Cy5 dyes' labeled DNA (not a single Cy5-labeled DNA). Can the authors show at least their diffusion time in solution is the same using FCS? Or a salt dependent measurement? Otherwise, the comment "minimum effect on its structure and motion under entangled conditions" is not valid yet.*

Thank you for the comment. From polymer physics point of view, structure (e.g. Kuhn length, entangled molecular weight, intrachain hydrodynamic interaction, etc.) and movement (e.g. motion at the scale of monomers, motion at the scale of entanglement segment, and motion at the scale of whole chain, etc.) of polymer chains under entangled conditions are coupled. Many previous studies on single-molecule entangled polymer dynamics of DNA (Manning GS, Q Rev. Biophys., 1987, 11, 179.; Perkins TT et al., Science, 1994, 264, 819.; Smith DE et al., Phys. Rev. Lett., 1995, 75, 4146.; Teixeira RE et al., Macromolecules, 2007, 40, 2461.; Robertson RM et al., Macromolecules, 2007, 40, 3373. etc.) used YOYO or TOTO dyes, which carry four positive charges per molecule. Observations in these studies strongly suggested that the fluorescence labeling of DNA with positively charged dyes has a minimum effect on the polymer dynamics, including static conformation as well as dynamic motion. Since the Cy5 dye carries one positive charge per molecule, in our opinion, it is relatively obvious that the labelling does not have a significant effect on the conformation and motion of the DNA molecules. Nevertheless, at the request of the reviewer, we conducted single-molecule tracking experiments on YOYO1-labelled and Cy5-labelled lambda DNA in a diluted solution. As can be seen in the below frequency histograms, we did not

observe any statistically significant differences in the diffusion coefficient between the two samples. This experiment demonstrated that the fluorescence labelling by Cy5 has a minimum effect on the motion of the DNA molecules.

3. *In Fig. 4, they used 9, 15, and 12 molecules for the analysis. I am not sure whether this small number is good enough to give a such a solid statistics. Can the authors provide a statistical justification on their analysis using such a small number of sampling?*

As the reviewer mentioned, we don't have a huge number of molecules to build the data displayed in Figure 4d, e, and f. Therefore, we carefully evaluated the position-dependent behaviours of the polymer chains based on statistics. As can be seen in these figures, the position dependences are much larger than error bars (that correspond to standard error). That means the observed position dependences are "statistically significant". Our conclusion is further supported by the fact that similar position dependences were not observed statistically in the entangled cyclic polymer chains. Together, we believe that we showed in a convincing way the position-dependent motion and relaxation of the entangled linear chains in this study.

Reviewer 2

1. *I am happy with the answers to my comments/questions. I was already in favor of publishing this manuscript, I am even more enthusiastic now.*

Thank you for the positive comment on our work.

REVIEWERS' COMMENTS:

Reviewer #1 (Remarks to the Author):

The authors answered my concerns well.

Point-by-point response to the referees

Reviewer 1

1. *The authors answered my concerns well.*

Thank you for the positive comment on our work.